# IQ-Learn: Inverse soft-Q Learning for Imitation

**Divyansh Garg**    **Shuvam Chakraborty**    **Chris Cundy**
**Jiaming Song**    **Stefano Ermon**
Stanford University
`{divgarg, shuvamc, cundy, tsong, ermon}@stanford.edu`

## Abstract

In many sequential decision-making problems (e.g., robotics control, game playing, sequential prediction), human or expert data is available containing useful information about the task. However, imitation learning (IL) from a small amount of expert data can be challenging in high-dimensional environments with complex dynamics. Behavioral cloning is a simple method that is widely used due to its simplicity of implementation and stable convergence but doesn't utilize any information involving the environment's dynamics. Many existing methods that exploit dynamics information are difficult to train in practice due to an adversarial optimization process over reward and policy approximators or biased, high variance gradient estimators. We introduce a method for dynamics-aware IL which avoids adversarial training by learning a single Q-function, implicitly representing both reward and policy. On standard benchmarks, the implicitly learned rewards show a high positive correlation with the ground-truth rewards, illustrating our method can also be used for inverse reinforcement learning (IRL). Our method, Inverse soft-Q learning (IQ-Learn) obtains state-of-the-art results in offline and online imitation learning settings, significantly outperforming existing methods both in the number of required environment interactions and scalability in high-dimensional spaces, often by more than 3x.

## 1   Introduction

Imitation of an expert has long been recognized as a powerful approach for sequential decision-making [29, 1], with applications as diverse as healthcare [39], autonomous driving [41], and playing complex strategic games [8]. In the imitation learning (IL) setting, we are given a set of expert trajectories, with the goal of learning a policy which induces behavior similar to the expert's. The learner has no access to the reward, and no explicit knowledge of the dynamics.

The simple behavioural cloning [34] approach simply maximizes the probability of the expert's actions under the learned policy, approaching the IL problem as a supervised learning problem. While this can work well in simple environments and with large quantities of data, it ignores the sequential nature of the decision-making problem, and small errors can quickly compound when the learned policy departs from the states observed under the expert. A natural way of introducing the environment dynamics is by framing the IL problem as an Inverse RL (IRL) problem, aiming to learn a reward function under which the expert's trajectory is optimal, and from which the learned imitation policy can be trained [1]. This framing has inspired several approaches which use rewards either explicitly or implicitly to incorporate dynamics while learning an imitation policy [17, 10, 33, 22]. However, these dynamics-aware methods are typically hard to put into practice due to unstable learning which can be sensitive to hyperparameter choice or minor implementation details [21].

In this work, we introduce a dynamics-aware imitation learning method which has stable, non-adversarial training, allowing us to achieve state-of-the-art performance on imitation learning bench-

35th Conference on Neural Information Processing Systems (NeurIPS 2021).

Table 1: A comparison of various algorithms for imitation learning. "Convergence Guarantees" refers to if a proof is given that the algorithm converges to the correct policy with sufficient data. We consider an algorithm "directly optimized" if it consists of an optimization algorithm (such as gradient descent) applied to the parameters of a single function

| | Method | Reference | Dynamics Aware | Non-Adversarial Training | Convergence Guarantees | Non-restrictive Reward | Direct Optimization |
|---|---|---|---|---|---|---|---|
| **Online** | Max Margin IRL | [29, 1] | ✓ | ✓ | ✓ | ✗ | ✗ |
| | Max Entropy IRL | [43] | ✓ | ✓ | ✓ | ✗ | ✗ |
| | GAIL/AIRL | [17, 10] | ✓ | ✗ | ✓ | ✓ | ✗ |
| | ASAF | [4] | ✓ | ✓ | ✓ | ✗ | ✓ |
| | SQIL | [33] | ✓ | ✓ | ✗ | ✗ | ✓ |
| | **Ours (Online)** | – | ✓ | ✓ | ✓ | ✓ | ✓ |
| **Offline** | Max Margin IRL | [24, 20] | ✓ | ✓ | ✓ | ✗ | ✗ |
| | Max Likelihood IRL | [18] | ✓ | ✓ | ✓ | ✗ | ✗ |
| | Max Entropy IRL | [16] | ✓ | ✓ | ✓ | ✗ | ✗ |
| | ValueDICE | [22] | ✓ | ✗ | ✗ | ✗ | ✗ |
| | Behavioral Cloning | [34] | ✗ | ✓ | ✓ | ✗ | ✓ |
| | Regularized BC | [30] | ✓ | ✓ | ✓ | ✗ | ✓ |
| | EDM | [19] | ✓ | ✓ | ✗ | ✓ | ✓ |
| | **Ours (Offline)** | – | ✓ | ✓ | ✓ | ✓ | ✓ |

marks. Our key insight is that much of the difficulty with previous IL methods arises from the IRL-motivated representation of the IL problem as a min-max problem over reward and policy [17, 1].

This introduces a requirement to separately model the reward and policy, and train these two functions jointly, often in an adversarial fashion. Drawing on connections between RL and energy-based models [13, 14], we propose learning a *single model for the Q-value*. The $Q$-value then implicitly defines both a reward and policy function. This turns a difficult min-max problem over policy and reward functions into a simpler minimization problem over a single function, the $Q$-value. Since our problem has a one-to-one correspondence with the min-max problem studied in adversarial IL [17], we maintain the generality and guarantees of these previous approaches, resulting in a meaningful reward that may be used for inverse reinforcement learning. Furthermore, our method may be used to minimize a variety of statistical divergences between the expert and learned policy. We show that we recover several previously-described approaches as special cases of particular divergences, such as the regularized behavioural cloning of [30], and the conservative Q-learning of [23].

In our experiments, we find that our method is performant even with very sparse data - surpassing prior methods using *one expert demonstration* in the completely offline setting - and can scale to complex image-based tasks like Atari reaching expert performance. Moreover, our learnt rewards are highly predictive of the original environment rewards.

Concretely, our contributions are as follows:

- We present a modified $Q$-learning update rule for imitation learning that can be implemented on top of soft-Q learning or soft actor-critic (SAC) algorithms in fewer than **15** lines of code.
- We introduce a simple framework to minimize a wide range of statistical distances: Integral Probability Metrics (IPMs) and f-divergences, between the expert and learned distributions.
- We empirically show state-of-art results in a variety of imitation learning settings: online and offline IL. On the complex Atari suite, we outperform prior methods by **3-7x** while requiring **3x** less environment steps.
- We characterize our learnt rewards and show a high positive correlation with the ground-truth rewards, justifying the use of our method for Inverse Reinforcement Learning.

## 2 Background

**Preliminaries** We consider environments represented as a Markov decision process (MDP), which is defined by a tuple $(\mathcal{S}, \mathcal{A}, p_0, \mathcal{P}, r, \gamma)$. $\mathcal{S}, \mathcal{A}$ represent state and action spaces, $p_0$ and $\mathcal{P}(s'|s, a)$ represent the initial state distribution and the dynamics, $r(s, a)$ represents the reward function, and

$\gamma \in (0, 1)$ represents the discount factor. $\mathbb{R}^{S \times A} = \{x : S \times A \to \mathbb{R}\}$ will denote the set of all functions in the state-action space and $\overline{\mathbb{R}}$ will denote the extended real numbers $\mathbb{R} \cup \{\infty\}$. Section 3 and 4 will work with finite state and action spaces $S$ and $A$, but our algorithms and experiments later in the paper use continuous environments. $\Pi$ is the set of all stationary stochastic policies that take actions in $A$ given states in $S$. We work in the $\gamma$-discounted infinite horizon setting, and we will use an expectation with respect to a policy $\pi \in \Pi$ to denote an expectation with respect to the trajectory it generates: $\mathbb{E}_\pi[r(s, a)] \triangleq \mathbb{E}[\sum_{t=0}^{\infty} \gamma^t r(s_t, a_t)]$, where $s_0 \sim p_0$, $a_t \sim \pi(\cdot|s_t)$, and $s_{t+1} \sim \mathcal{P}(\cdot|s_t, a_t)$ for $t \geq 0$. For a policy $\pi \in \Pi$, we define its occupancy measure $\rho_\pi : S \times A \to \mathbb{R}$ as $\rho_\pi(s, a) = \pi(a|s) \sum_{t=0}^{\infty} \gamma^t P(s_t = s|\pi)$. We refer to the expert policy as $\pi_E$ and its occupancy measure as $\rho_E$. In practice, $\pi_E$ is unknown and we have access to a sampled dataset of demonstrations. For brevity, we refer to $\rho_\pi$ as $\rho$ for a learnt policy in the paper.

**Soft $Q$-functions** For a reward $r \in \mathbb{R}^{S \times A}$ and $\pi \in \Pi$, the soft Bellman operator $\mathcal{B}^\pi$ : $\mathbb{R}^{S \times A} \to \mathbb{R}^{S \times A}$ defined as $(\mathcal{B}^\pi Q)(s, a) = r(s, a) + \gamma \mathbb{E}_{s' \sim P(s, a)} V^\pi(s')$ with $V^\pi(s) = \mathbb{E}_{a \sim \pi(\cdot|s)}[Q(s, a) - \log \pi(a|s)]$. The soft Bellman operator is contractive [13] and defines a unique soft $Q$-function for $r$, given as $Q = \mathcal{B}^\pi Q$.

**Max Entropy Reinforcement Learning** For a given reward function $r \in \mathbb{R}^{S \times A}$, maximum entropy RL [14, 5] aims to learn a policy that maximizes the expected cumulative discounted reward along with the entropy in each state: $\max_{\pi \in \Pi} \mathbb{E}_\pi[r(s, a)] + H(\pi)$. Where $H(\pi) \triangleq \mathbb{E}_\pi[- \log \pi(a|s)]$ is the discounted causal entropy of the policy $\pi$. The optimal policy satisfies [42, 5]:

$$\pi^*(a|s) = \frac{1}{Z_s} \exp(Q(s, a)), \tag{1}$$

where $Q$ is the soft $Q$-function and $Z_s$ is the normalization factor given as $\sum_{a'} \exp(Q(s, a'))$.

$Q$ satisfies the soft-Bellman equation:

$$Q(s, a) = r(s, a) + \gamma \mathbb{E}_{s' \sim \mathcal{P}(\cdot|s, a)} \left[ \log \sum_{a'} \exp(Q(s', a')) \right] \tag{2}$$

In continuous action spaces, $Z_s$ becomes intractable and soft actor-critic methods like SAC [13] can be used to learn an explicit policy.

**Max Entropy Inverse Reinforcement Learning** Given demonstrations sampled using the policy $\pi_E$, maximum entropy Inverse RL aims to recover the reward function in a family of functions $\mathcal{R}$ that rationalizes the expert behavior by solving the optimization problem: $\max_{r \in \mathcal{R}} \min_{\pi \in \Pi} \mathbb{E}_{\pi_E}[r(s, a)] - (\mathbb{E}_\pi[r(s, a)] + H(\pi))$, where the expected reward of $\pi_E$ is empirically approximated. It looks for a reward function that assigns high reward to the expert policy and a low reward to other policies, while searching for the best policy for the reward function in an inner loop.

The Inverse RL objective can be reformulated in terms of its occupancy measure, and with a convex reward regularizer $\psi : \mathbb{R}^{S \times A} \to \overline{\mathbb{R}}$ [17]

$$\max_{r \in \mathcal{R}} \min_{\pi \in \Pi} L(\pi, r) = \mathbb{E}_{\rho_E}[r(s, a)] - \mathbb{E}_\rho[r(s, a)] - H(\pi) - \psi(r) \tag{3}$$

In general, we can exchange the max-min resulting in an objective that minimizes the statistical distance parameterized by $\psi$, between the expert and the policy [17]

$$\min_{\pi \in \Pi} \max_{r \in \mathcal{R}} L(\pi, r) = \min_{\pi \in \Pi} d_\psi(\rho, \rho_E) - H(\pi), \tag{4}$$

with $d_\psi \triangleq \psi^*(\rho_E - \rho)$, where $\psi^*$ is the convex conjugate of $\psi$.

## 3 Inverse soft Q-learning (IQ-Learn) Framework

A naive solution to the IRL problem in (Eq. 3) involves (1) an outer loop learning rewards and (2) executing RL in an inner loop to find an optimal policy for them. However, we know that this optimal policy can be obtained analytically in terms of soft $Q$-functions (Eq. 1). Interestingly, as we will show

later, the rewards can also be represented in terms of $Q$ (Eq. 2). Together, these observations suggest it might be possible to directly solve the IRL problem by optimizing only over the $Q$-function.

To motivate the search of an imitation learning algorithm that depends only on the $Q$-function, we characterize the space of $Q$-functions and policies obtained using Inverse RL. We will study $\pi \in \Pi$, $r \in \mathcal{R}$ and $Q$-functions $Q \in \Omega$ where $\mathcal{R} = \Omega = \mathbb{R}^{\mathcal{S} \times \mathcal{A}}$. We assume $\Pi$ is convex, compact and that $\pi_E \in \Pi$[1]. We define $V^\pi(s) = \mathbb{E}_{a \sim \pi(\cdot|s)}[Q(s,a) - \log \pi(a|s)]$.

We start with analysis developed in [17]: The regularized IRL objective $L(\pi, r)$ given by Eq. 3, is concave in the policy and convex in rewards. And has a unique saddle point where it is optimized.

To characterize the $Q$-functions it is useful to transform the optimization problem over rewards to a problem over $Q$-functions. We can get a one-to-one correspondence between $r$ and $Q$:

Define the inverse soft bellman operator $\mathcal{T}^\pi : \mathbb{R}^{\mathcal{S} \times \mathcal{A}} \to \mathbb{R}^{\mathcal{S} \times \mathcal{A}}$ such that
$$(\mathcal{T}^\pi Q)(s,a) = Q(s,a) - \gamma \mathbb{E}_{s' \sim P(s,a)} V^\pi(s'),$$

**Lemma 3.1.** *The inverse soft bellman operator $\mathcal{T}^\pi$ is bijective, and for any r, $(\mathcal{T}^\pi)^{-1} r$ is the unique contraction of $\mathcal{B}^\pi$.*

The proof of this lemma is in Appendix A.1. For a policy $\pi$, we are thus justified in changing between rewards and their corresponding soft-Q functions. We can freely transform functions from the reward-policy space: $\Pi \times \mathcal{R}$ to the $Q$-policy space: $\Pi \times \Omega$, giving us the lemma:

**Lemma 3.2.** *If $L(\pi, r) = \mathbb{E}_{\rho_E}[r(s,a)] - \mathbb{E}_\rho[r(s,a)] - H(\pi) - \psi(r)$ and $\mathcal{J}(\pi, Q) = \mathbb{E}_{\rho_E}[(\mathcal{T}^\pi Q)(s,a)] - \mathbb{E}_\rho[(\mathcal{T}^\pi Q)(s,a)] - H(\pi) - \psi(\mathcal{T}^\pi Q)$, then for all policies $\pi \in \Pi$, $L(\pi, r) = \mathcal{J}(\pi, (\mathcal{T}^\pi)^{-1} r)$ for all $r \in \mathcal{R}$, and $\mathcal{J}(\pi, Q) = L(\pi, \mathcal{T}^\pi Q)$, for all $Q \in \Omega$.*

Lemma 3.1 and 3.2 allow us to adapt the Inverse RL objective $L(\pi, r)$ to learning $Q$ through $\mathcal{J}(\pi, Q)$.

Simplifying our new objective (using Lemma A.3 in Appendix):
$$\mathcal{J}(\pi, Q) = \mathbb{E}_{s,a \sim \rho_E}[Q - \gamma \mathbb{E}_{s' \sim \mathcal{P}(\cdot|s,a)} V^\pi(s')] - (1-\gamma)\mathbb{E}_{s_0 \sim p_0}[V^\pi(s_0)] - \psi(\mathcal{T}^\pi Q), \quad (5)$$

We are now ready to study $\mathcal{J}(\pi, Q)$, the Inverse RL optimization problem in the $Q$-policy space. As the regularizer $\psi$ depends on both $Q$ and $\pi$, a general analysis over all functions in $\mathbb{R}^{\mathcal{S} \times \mathcal{A}}$ becomes too difficult. We restrict ourselves to regularizers induced by a convex function $g : \mathbb{R} \to \overline{\mathbb{R}}$ such that
$$\psi_g(r) = \mathbb{E}_{\rho_E}[g(r(s,a))] \qquad (6)$$

This allows us to simplify our analysis to the set of all real functions while retaining generality[2]. We further motivate this choice in Section 4.

**Proposition 3.3.** *In the Q-policy space, there exists a unique saddle point $(\pi^*, Q^*)$ that optimizes $\mathcal{J}$. i.e. $Q^* = \mathrm{argmax}_{Q \in \Omega} \min_{\pi \in \Pi} \mathcal{J}(\pi, Q)$ and $\pi^* = \mathrm{argmin}_{\pi \in \Pi} \max_{Q \in \Omega} \mathcal{J}(\pi, Q)$. Furthermore, $\pi^*$ and $r^* = \mathcal{T}^{\pi^*} Q^*$ are the solution to the Inverse RL objective $L(\pi, r)$.*

Thus we have, $\max_{Q \in \Omega} \min_{\pi \in \Pi} \mathcal{J}(\pi, Q) = \max_{r \in \mathcal{R}} \min_{\pi \in \Pi} L(\pi, r)$.

This tells us, even after transforming to $Q$-functions we have retained the saddle point property of the original IRL objective and optimizing $\mathcal{J}(\pi, Q)$ recovers this saddle point. In the $Q$-policy space, we can get an additional property:

**Proposition 3.4.** *For a fixed Q, $\mathrm{argmin}_{\pi \in \Pi} \mathcal{J}(\pi, Q)$ is the solution to max entropy RL with rewards $r = \mathcal{T}^\pi Q$. Thus, this forms a manifold in the Q-policy space, that satisfies*
$$\pi_Q(a|s) = \frac{1}{Z_s} \exp(Q(s,a)),$$
*with normalization factor $Z_s = \sum_a \exp Q(s,a)$ and $\pi_Q$ defined as the $\pi$ corresponding to Q.*

Proposition 3.3 and 3.4 are telling us that if we know $Q$, then the inner optimization problem in terms of policy is trivial, and obtained in a closed form! Thus, we can recover an objective that only requires learning $Q$:
$$\max_{Q \in \Omega} \min_{\pi \in \Pi} \mathcal{J}(\pi, Q) = \max_{Q \in \Omega} \mathcal{J}(\pi_Q, Q) \qquad (7)$$

Furthermore, we have:

---

[1]The full policy class satisfies all these assumptions

[2]Averaging over the expert occupancy allows $\psi$ to adjust to arbitrary experts and accommodate multimodality

**Proposition 3.5.** *Let $\mathcal{J}^*(Q) = \mathcal{J}(\pi_Q, Q)$. Then $\mathcal{J}^*$ is concave in $Q$.*

Thus, this new optimization objective is well-behaved and is maximized only at the saddle point.

In Appendix C, we expand on our analysis and characterize the behavior for different choices of regularizer $\psi$, while giving proofs of all our propositions. Figure 1 summarizes the properties for the IRL objective: there exists a optimal policy manifold depending on $Q$, allowing optimization along it (using $\mathcal{J}^*$) to converge to

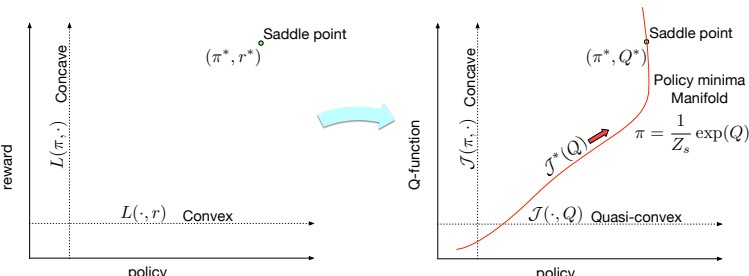

Figure 1: Properties of IRL objective in reward-policy space and Q-policy space.

the saddle point. We further present analysis of IL methods that learn $Q$-functions like SQIL [33] and ValueDICE [22] and find subtle fallacies affecting their learning.

Note that although the same analysis holds in the reward-policy space, the optimal policy manifold depends on $Q$, which isn't trivially known unlike when in the Q-policy space.

## 4 Approach

In this section, we develop our inverse soft-Q learning (IQ-Learn) algorithm, such that it recovers the optimal soft $Q$-function for a MDP from a given expert distribution. We start by learning energy-based models for the policy similar to soft $Q$-learning and later learn an explicit policy similar to actor-critic methods.

### 4.1 General Inverse RL Objective

For designing a practical algorithm using regularizers of the form $\psi_g$ (from Eq. 6), we define $g$ using a concave function $\phi : \mathcal{R}_\psi \to \mathbb{R}$, such that $\quad g(x) = \begin{cases} x - \phi(x) & \text{if } x \in \mathcal{R}_\psi \\ +\infty & \text{otherwise} \end{cases}$

with the rewards constrained in $R_\psi$.

For this choice of $\psi$, the Inverse RL objective $L(\pi, r)$ takes the form of Eq. 4 with a distance measure:

$$d_\psi(\rho, \rho_E) = \max_{r \in \mathcal{R}_\psi} \mathbb{E}_{\rho_E}[\phi(r(s, a))] - \mathbb{E}_\rho[r(s, a)], \tag{8}$$

This forms a general learning objective that allows the use of a wide-range of statistical distances including Integral Probability Metrics (IPMs) and f-divergences (see Appendix B). [3]

### 4.2 Choice of Statistical Distances

While choosing a practical regularizer, it can be useful to obtain certain properties on the reward functions we recover. Some (natural) nice properties are: having rewards bounded in a range, learning smooth functions or enforcing a norm-penalty.

In fact, we find these properties correspond to the Total Variation distance, the Wasserstein-1 distance and the $\chi^2$-divergence respectively. The regularizers and the induced statistical distances are summarized in Table 2:

Table 2: Enforced reward property, corresponding regularizer $\psi$ and statistical distance ($R_{\max}, K, \alpha \in \mathbb{R}^+$)

| Reward Property | $\psi$ | $d_\psi$ |
|---|---|---|
| Bound range | $\psi = 0$ if $|r| \leq R_{\max}$ and $+\infty$ otherwise | $2R_{\max} \cdot \text{TV}(\rho, \rho_E)$ |
| Smoothness | $\psi = 0$ if $\|r\|_{\text{Lip}} \leq K$ and $+\infty$ otherwise | $K \cdot W_1(\rho, \rho_E)$ |
| L2 Penalization | $\psi(r) = \alpha r^2$ | $\frac{1}{4\alpha} \cdot \chi^2(\rho, \rho_E)$ |

[3]We recover IPMs when using identity $\phi$ and restricted reward family $\mathcal{R}$

We find that these choice of regularizers[4] work very well in our experiments. In Appendix B, we further give a table for the well known $f$-divergences, the corresponding $\phi$ and the learnt reward estimators, along with a result ablation on using different divergences. Compared to $\chi^2$, we find other $f$-divergences like Jensen-Shannon result in similar performances but are not as readily interpretable.

### 4.3 Inverse soft-Q update (Discrete control)

Optimization along the optimal policy manifold gives the concave objective (Prop 3.5):

$$\max_{Q \in \Omega} \mathcal{J}^*(Q) = \mathbb{E}_{\rho_E}[\phi(Q(s,a) - \gamma \mathbb{E}_{s' \sim \mathcal{P}(\cdot|s,a)} V^*(s'))] - (1-\gamma)\mathbb{E}_{\rho_0}[V^*(s_0)], \qquad (9)$$

with $V^*(s) = \log \sum_a \exp Q(s,a)$.

For each $Q$, we get a corresponding reward $r(s,a) = Q(s,a) - \gamma \mathbb{E}_{s' \sim \mathcal{P}(\cdot|s,a)}[\log \sum_{a'} \exp Q(s',a')]$. This correspondence is unique (Lemma A.1 in Appendix), and every update step can be seen as finding a better reward for IRL.

Note that estimating $V^*(s)$ exactly is only possible in discrete action spaces. Our objective forms a variant of soft-Q learning: to learn the optimal $Q$-function given an expert distribution.

### 4.4 Inverse soft actor-critic update (Continuous control)

In continuous action spaces, it might not be possible to exactly obtain the optimal policy $\pi_Q$, which forms an energy-based model of the $Q$-function, and we use an explicit policy $\pi$ to approximate $\pi_Q$.

For any policy $\pi$, we have a objective (from Eq. 5):

$$\mathcal{J}(\pi, Q) = \mathbb{E}_{\rho_E}[\phi(Q - \gamma \mathbb{E}_{s' \sim \mathcal{P}(\cdot|s,a)} V^\pi(s'))] - (1-\gamma)\mathbb{E}_{\rho_0}[V^\pi(s_0)] \qquad (10)$$

For a fixed $Q$, soft actor-critic (SAC) update: $\min_\pi \mathbb{E}_{s \sim \mathcal{D}, a \sim \pi(\cdot|s)}[Q(s,a) - \log \pi(a|s)]$, brings $\pi$ closer to $\pi_Q$ while always minimizing Eq. 10 (Lemma A.4 in Appendix). Here $\mathcal{D}$ is the distribution of previously sampled states, or a replay buffer.

Thus, we obtain the modified actor-critic update rule to learn $Q$-functions from the expert distribution:

1. For a fixed $\pi$, optimize $Q$ by maximizing $\mathcal{J}(\pi, Q)$.
2. For a fixed $Q$, apply SAC update to optimize $\pi$ towards $\pi_Q$.

This differs from ValueDICE [22], where the actor is updated adversarially and the objective may not always converge (Appendix C).

## 5 Practical Algorithm

Pseudocode in Algorithm 1, shows our $Q$-learning and actor-critic variants, with differences with conventional RL algorithms in red (we optimize $-\mathcal{J}$ to use gradient descent). We can implement our algorithm IQ-Learn in **15** lines of code on top of standard implementations of (soft) DQN [14] for discrete control or soft actor-critic (SAC) [13] for continuous control, with a change on the objective for the $Q$-function. Default hyperparameters from [14, 13] work well, except for tuning the entropy regularization. Target networks were helpful for continuous control. We elaborate details in Appendix D.

### 5.1 Training methodology

Corollary 2.1 in Appendix A states $\mathbb{E}_{(s,a) \sim \mu}[V^\pi(s) - \gamma \mathbb{E}_{s' \sim \mathcal{P}(\cdot|s,a)} V^\pi(s')] = (1-\gamma)\mathbb{E}_{s \sim p_0}[V^\pi(s)]$, where $\mu$ is any policy's occupancy. We use this to stabilize training instead of using Eq. 9 directly.

**Online**: Instead of directly estimating $\mathbb{E}_{p_0}[V^\pi(s_0)]$ in our algorithm, we can sample $(s,a,s')$ from a replay buffer and get a single-sample estimate $\mathbb{E}_{(s,a,s') \sim \text{replay}}[V^\pi(s) - \gamma V^\pi(s')]$. This removes the issue where we are only optimizing $Q$ in the inital states resulting in overfitting of $V^\pi(s_0)$, and improves the stability for convergence in our experiments. We find sampling half from the policy buffer and half from the expert distribution gives the best performances. Note that this is makes our learning online, requiring environment interactions.

---

[4]The additional scalar terms scale the entropy regularization strength and can be ignored in practice

**Offline**: Although $\mathbb{E}_{p_0}[V^\pi(s_0)]$ can be estimated offline we still observe an overfitting issue. Instead of requiring policy samples we use only expert samples to estimate $\mathbb{E}_{(s,a,s')\sim\text{expert}}[V^\pi(s) - \gamma V^\pi(s')]$ to sufficiently approximate the term. This methodology gives us state-of-art results for offline IL.

---

**Algorithm 1** Inverse soft Q-Learning (both variants)

1: Initialize Q-function $Q_\theta$, and optionally a policy $\pi_\phi$
2: **for** step $t$ in $\{1...N\}$ **do**
3:     Train Q-function using objective from Equation 9:
    $\theta_{t+1} \leftarrow \theta_t - \alpha_Q \nabla_\theta[-\mathcal{J}(\theta)]$
    (Use $V^*$ for Q-learning and $V^{\pi_\phi}$ for actor-critic)
4:     (only with actor-critic) Improve policy $\pi_\phi$ with SAC style actor update:
    $\phi_{t+1} \leftarrow \phi_t - \alpha_\pi \nabla_\phi \mathbb{E}_{s\sim\mathcal{D}, a\sim\pi_\phi(\cdot|s)}[Q(s,a) - \log \pi_\phi(a|s)]$
5: **end for**

---

**Algorithm 2** Recover policy and reward

1: Given trained Q-function $Q_\theta$, and optionally a trained policy $\pi_\phi$
2: Recover policy $\pi$:
    (Q-learning) $\pi := \frac{1}{Z}\exp Q_\theta$
    (actor-critic) $\pi := \pi_\phi$
3: For state $\mathbf{s}$, action $\mathbf{a}$ and $\mathbf{s}' \sim \mathcal{P}(\cdot|\mathbf{s},\mathbf{a})$
4: Recover reward $r(\mathbf{s},\mathbf{a},\mathbf{s}') = Q_\theta(\mathbf{s},\mathbf{a}) - \gamma V^\pi(\mathbf{s}')$

---

### 5.2 Recovering rewards

Instead of the conventional reward function $r(s,a)$ on state and action pairs, our algorithm allows recovering rewards for each transition $(s,a,s')$ using the learnt $Q$-values as follows:

$$r(s,a,s') = Q(s,a) - \gamma V^\pi(s') \tag{11}$$

Now, $\mathbb{E}_{s'\sim\mathcal{P}(\cdot|s,a)}[Q(s,a) - \gamma V^\pi(s')] = Q(s,a) - \gamma\mathbb{E}_{s'\sim\mathcal{P}(\cdot|s,a)}[V^\pi(s')] = \mathcal{T}^\pi Q(s,a)$. This is just the reward function $r(s,a)$ we want. So by marginalizing over next-states, our expression correctly recovers the reward over state-actions. Thus, Eq. 11 gives the reward over transitions.

Our rewards require $s'$ which can be sampled from the environment, or by using a dynamics model.

### 5.3 Implementation of Statistical Distances

Implementing TV and $W_1$ distances is fairly trivial and we give details in Appendix B. For the $\chi^2$-**divergence**, we note that it corresponds to $\phi(x) = x - \frac{1}{4\alpha}x^2$. On substituting in Eq. 9, we get

$$\max_{Q\in\Omega} \mathbb{E}_{\rho_E}[(Q(s,a) - \gamma\mathbb{E}_{s'\sim\mathcal{P}(\cdot|s,a)}V^*(s'))] - (1-\gamma)\mathbb{E}_{p_0}[V^*(s_0)] - \frac{1}{4\alpha}\mathbb{E}_{\rho_E}[(Q(s,a) - \gamma\mathbb{E}_{s'\sim\mathcal{P}(\cdot|s,a)}V^*(s'))^2]$$

In a fully offline setting, this can be further simplified as (using the offline methodology in Sec 5.1):

$$\min_{Q\in\Omega} -\mathbb{E}_{\rho_E}[(Q(s,a) - V^*(s))] + \frac{1}{4\alpha}\mathbb{E}_{\rho_E}[(Q(s,a) - \gamma\mathbb{E}_{s'\sim\mathcal{P}(\cdot|s,a)}V^*(s'))^2] \tag{12}$$

This is interestingly the same as the $Q$-learning objective in CQL [23], an state-of-art method for offline RL (using 0 rewards), and shares similarities with regularized behavior cloning [33] [5].

### 5.4 Learning state-only reward functions

Previous works like AIRL [10] propose learning rewards that are only function of the state, and claim that these form of reward functions generalize between different MDPs. We find our method can predict state-only rewards by using the policy and expert state-marginals with a modification to Eq. 9:

$$\max_{Q\in\Omega} \mathcal{J}^*(Q) = \mathbb{E}_{s\sim\rho_E(s)}[\mathbb{E}_{a\sim\pi(\cdot|s)}[\phi(Q(s,a) - \gamma\mathbb{E}_{s'\sim\mathcal{P}(\cdot|s,a)}V^*(s'))]] - (1-\gamma)\mathbb{E}_{p_0}[V^*(s_0)]$$

Interestingly, our objective no longer depends on the the expert actions $\pi_E$ and can be used for IL using only observations. For the sake of brevity, we expand on this in Section 1 in Appendix A.

## 6 Related Work

**Classical IL**: Imitation learning has a long history, with early works using supervised learning to match a policy's actions to those of the expert [15, 35]. A significant advance was made with the formulation of IL as the composition of RL and IRL [29, 1, 43], recovering the expert's policy

---

[5]The simplification to get Eq. (12) is not applicable in the online IL setting where our method differs

by inferring the expert's reward function, then finding the policy which maximizes reward under this reward function. These early approaches required a hand-designed featurization of the MDP, limiting their applicability to complex MDPs. In this setting, early approaches [9, 31] noted a formal equivalence between IRL and IL using an inverse Bellman operator similar to our own.

**Online IL**: More recent work aims to leverage the power of modern machine learning approaches to learn good featurizations and extend IL to complex settings. Recent work generally falls into one of two settings: online or offline. In the online setting, the IL algorithm is able to interact with the environment to obtain dynamics information. GAIL [17] takes the nested RL/IRL formulation of earlier work , optimizing over all reward functions with a convex regularizer. This results in the objective in Eq. (3), with a max-min adversarial problem similar to a GAN [11]. A variety of further work has built on this adversarial approach [21, 10, 3]. A separate line of work aims to simplify the problem in Eq. (3) by using a fixed $r$ or $\pi$. In SQIL [33], $r$ is chosen to be the 1-0 indicator on the expert demonstrations, while ASAF [4] takes the GAN approach and uses a discriminator (with role similar to $r$) of fixed form, consisting of a ratio of expert and learner densities. AdRIL [38] is a recent extension of SQIL, additionally assigning decaying negative reward to previous policy rollouts.

**Offline IL**: In the offline setting, the learner has no access to the environment. The simple behavioural cloning (BC) [34] approach is offline, but doesn't use any dynamics information. ValueDICE [22] is a dynamics-aware offline approach with an objective somewhat similar to ours, motivated from minimization of a variational representation of the KL-divergence between expert and learner policies. ValueDICE requires adversarial optimization to learn the policy and Q-functions, with a biased gradient estimator for training. We show a way to recover a unbiased gradient estimate for the KL-divergence in Appendix C. The O-NAIL algorithm [2] builds on ValueDICE and combines with an SAC update to obtain a method that is similar to our algorithm described in section 4.4, with the specific choice of reverse KL-divergence as the relevant statistical distance. The EDM method [19] incorporates dynamics via learning an explicit energy based model for the expert state occupancy, although some theoretical details have been called into question (see [37] for details). The recent AVRIL approach [6] uses a variational method to solve a probabilistic formulation of IL, finding a posterior distribution over $r$ and $\pi$. Illustrating the potential benefits of alternative distances for IL, the PWIL [7] algorithm gives a non-adversarial procedure to minimize the Wasserstein distance between expert and learned occupancies. The approach is specific to the primal form of the $\mathcal{W}_1$-distance, while our method (when used with the Wasserstein distance) targets the dual form.

# 7 Experiments

## 7.1 Experimental Setup

We compare IQ-Learn ("IQ") to prior work on a diverse collection of RL tasks and environments - ranging from low-dimensional control tasks: CartPole, Acrobot, LunarLander - to more challenging continuous control MuJoCo tasks: HalfCheetah, Hopper, Walker and Ant. Furthermore, we test on the visually challenging Atari Suite with high-dimensional image inputs. We compare on offline IL - with no access to the the environment while training, and online IL - with environment access. We show results on $W_1$ and $\chi^2$ as our statistical distances, as we found them more effective than TV distance. In all cases, we train until convergence and average over multiple seeds. Hyperparameter settings and training details are detailed in Appendix D.

## 7.2 Benchmarks

**Offline IL**   We compare to the state-of-art IL methods EDM and AVRIL, following the same experimental setting as [6]. Furthermore, we compare with ValueDICE which also learns Q-functions, albeit with drawbacks such as adversarial optimization. We also experimented with SQIL, but found that it was not competitive in the offline setting. Finally, we utilize BC as an additional IL baseline.

**Online IL**   We use MuJoCo and Atari environments and compare against state-of-art online IL methods: ValueDICE, SQIL and GAIL. We only show results on $\chi^2$ as $W_1$ was harder to stabilize on complex environments[6]. Using target updates stabilizes the $Q$-learning on MuJoCo. For brevity, further online IL results are shown in the Appendix D.

---

[6]$\chi^2$ and $W_1$ can be used together to still have a convex regularization and is more stable

## 7.3  Results

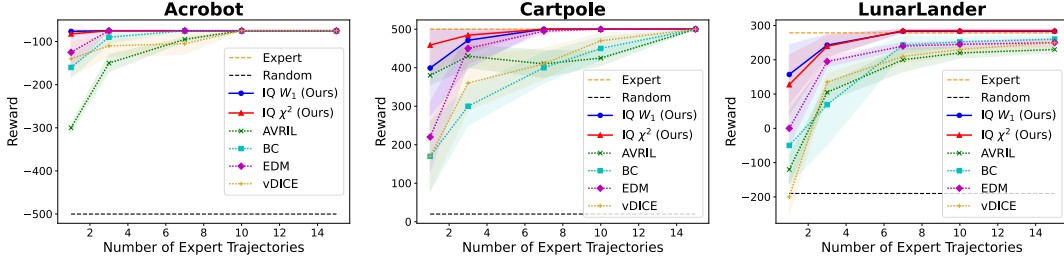

Figure 2: **Offline IL results.** We plot the average environment returns vs the number of expert trajectories.

**Offline IL**   We present results on the three offline control tasks in Figure 2. On all tasks, IQ strongly outperforms prior works we compare to in performance and sample efficiency. Using just *one expert trajectory*, we achieve expert performance on Acrobot and reach near expert on Cartpole.

**Mujoco Control**   We present our results on the MuJoCo tasks using a single expert demo in Table 3. IQ achieves expert-level performance in all the tasks while outperforming prior methods like ValueDICE and GAIL. We did not find SQIL competitive in this setting, and skip it for brevity.

Table 3: **Mujoco Results.** We show our performance on MuJoCo control tasks using a single expert trajectory.

| Task | GAIL | DAC | ValueDICE | IQ (Ours) | Expert |
|---|---|---|---|---|---|
| Hopper | 3252.5 | 3305.1 | 3312.1 | **3546.4** | 3532.7 |
| Half-Cheetah | 3080.0 | 4080.6 | 3835.6 | **5076.6** | 5098.3 |
| Walker | 4013.7 | 4107.9 | 3842.6 | **5134.0** | 5274.5 |
| Ant | 2299.1 | 1437.5 | 1806.3 | **4362.9** | 4700.0 |
| Humanoid | 232.6 | 380.5 | 644.5 | **5227.1** | 5312.8 |

**Atari**   We present our results on Atari using 20 expert demos in Figure 3. We reach expert performance on Space Invaders while being near expert on Pong and Breakout. Compared to prior methods like SQIL, IQ obtains **3-7x** normalized score[7] and converges in ∼300k steps, being **3x** faster compared to Q-learning based RL methods that take more than 1M steps to converge. Other popular methods like GAIL and ValueDICE perform near random even with 1M env steps.

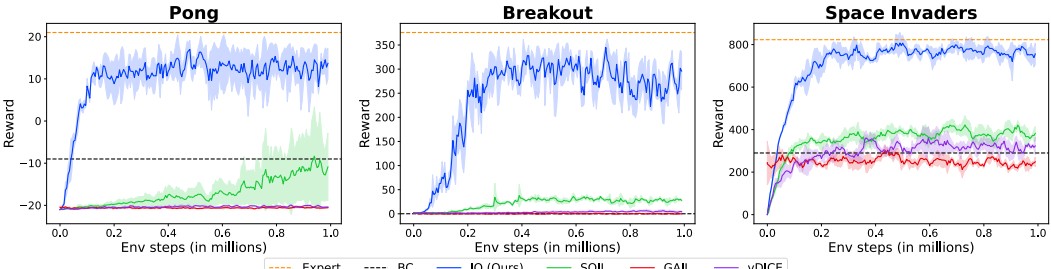

Figure 3: **Atari Results**. We show the returns vs the number of env steps. (Averaged over 5 seeds)

## 7.4  Recovered Rewards

IQ has the added benefit of recovering rewards and can be used for IRL. On Hopper task, our learned rewards have a Pearson correlation of **0.99** with the true rewards. In Figure 4, we visualize our recovered rewards in a simple grid environment. We elaborate details in Appendix D.

## 8  Discussion and Outlook

We present a new principled framework for learning soft-$Q$ functions for IL and recovering the optimal policy and the reward, building on past works in IRL [43]. Our algorithm IQ-Learn outperforms prior methods with very sparse expert data and scales to complex image-based environments. We also recover rewards highly correlated with actual rewards. It has applications in autonomous driving and complex decision-making, but proper considerations need to be taken into account to ensure safety

---

[7]normalizing rewards obtained from random behavior to 0 and expert to 1

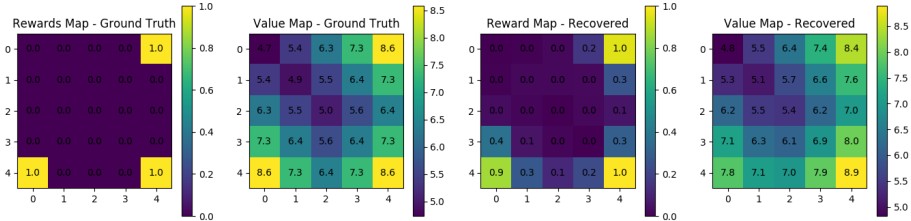

Figure 4: **Reward Visualization.** We use a discrete GridWorld environment with 5 possible actions: up, down, left, right, stay. Agent starts in a random state. (With 30 expert demos)

and reduce uncertainty, before any deployment. Finally, human or expert data can have errors that can propagate. A limitation of our method is that our recovered rewards depend on the environment dynamics, preventing trivial use on reward transfer settings. One direction of future work could be to learn a reward model from the trained soft-$Q$ model to make the rewards explicit.

# 9 Acknowledgements

We thank Kuno Kim and John Schulman for helpful discussions. We also thank Ian Goodfellow as some initial motivations for this work were developed under an internship with him.

# 10 Funding Transparency

This research was supported in part by NSF (#1651565, #1522054, #1733686), ONR (N00014-19-1-2145), AFOSR (FA9550-19-1-0024) and FLI.

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
