# A Appendix A

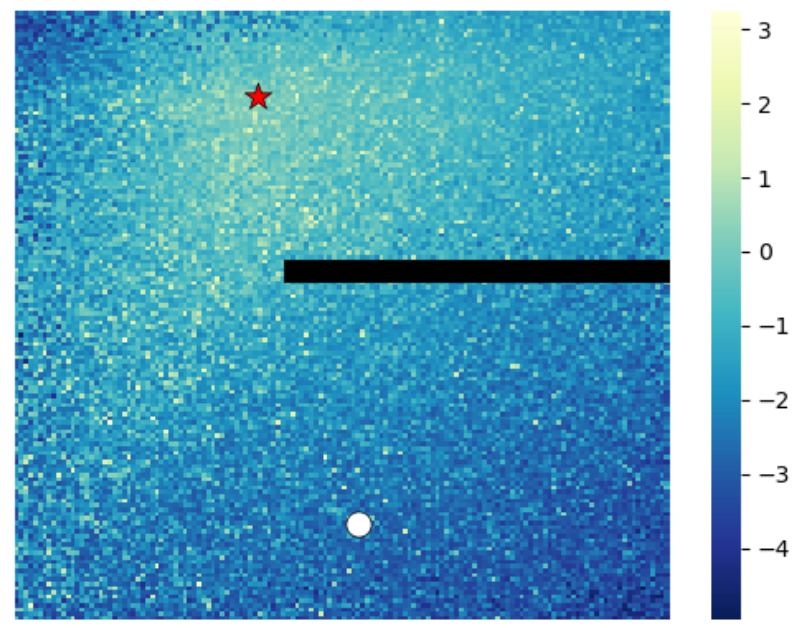

Figure 5: **State Rewards Visualization.** We visualize the state-only rewards recovered on a continuous control point maze task. The agent (white circle) has to reach the goal (red star) avoiding the barrier on right.

## A.1 Learning with state-only rewards

For a policy $\pi \in \Pi$, we define its state-marginal occupancy measure $\rho_\pi : \mathcal{S} \to \mathbb{R}$ as $\rho_\pi(s) = \sum_{t=0}^{\infty} \gamma^t P(s_t = s|\pi)$.

Suppose we are interested in learning rewards that are functions of only the states, then the Inverse-RL objective $L$ from Eq. 3 becomes a function of the state-marginal occupancies:

$$\max_{r \in \mathcal{R}_\psi} \min_{\pi \in \Pi} L_s(\pi, r) = \mathbb{E}_{s \sim \rho_E(s)}[\phi(r(s))] - \mathbb{E}_{s \sim \rho(s)}[r(s)] - H(\pi) \tag{13}$$

Now, we can parameterize the rewards $r(s)$ using state-only value-functions $V(s)$ and remove the dependency on $Q(s, a)$. Then $V(s)$ can be learnt similar to learning $Q(s, a)$ in the main paper, but $Q(s, a)$ remains unknown and the optimal policy cannot be obtained simply as an energy-based model of $Q$.

Instead, we develop a new objective that can learn $Q$ while recovering state-only rewards below.

We expand the original objective $L$ using the expert occupancy:

$$L(\pi, r) = \mathbb{E}_{s \sim \rho_E(s)} \mathbb{E}_{a \sim \pi_E(\cdot|s)} \left[ \phi(r(s, a)) - \frac{\rho(s)\pi(a|s)}{\rho_E(s)\pi_E(a|s)} r(s, a)) \right] - H(\pi) \tag{14}$$

We see that the action dependency comes in the equation from the fact that we have $\pi/\pi_E$ inside.

Now, we propose to fix the expression to make it independent of actions by replacing the expert policy $\pi_E$ with the policy $\pi$. The new objective becomes:

$$L'(\pi, r) = \mathbb{E}_{s \sim \rho_E(s)} \mathbb{E}_{a \sim \pi(\cdot|s)} \left[ \phi(r(s, a)) - \frac{\rho(s)}{\rho_E(s)} r(s, a) \right] - H(\pi) \tag{15}$$

Then for a fixed policy $\pi$, while maximizing over $r$ the constraint we have is that each reward component $r(s, a) \in R_\psi$. In a state $s$, $r(s, a)$ that maximizes the objective will take the same value independent of the action[8]. Thus, the expectation over actions can be removed and this recovers Eq. 13.

Writing the new objective using $Q$-functions, we get the modification to Eq. 9:

$$\max_{Q \in \Omega} \mathcal{J}^*(Q) = \mathbb{E}_{s \sim \rho_E(s)}[\mathbb{E}_{a \sim \pi(\cdot|s)}[\phi(Q(s, a) - \gamma \mathbb{E}_{s' \sim P(s,a)} V^*(s'))]] - (1 - \gamma)\mathbb{E}_{p_0}[V^*(s_0)] \tag{16}$$

This new objective does not depend on the the expert actions $\pi_E$ and can be used for IL using only observations (ILO). We visualize state-only rewards recovered on a 2D point mass navigation task in Fig 5. Notice that the rewards are not directional and are high on all sides of the target point, indicating they are not dependent on the action. We present additional results in Appendix D.

[8]The objective and the reward constraints remain same along each action dimension and a symmetry argument holds

### A.2 Proofs for Section 3 and Section 4

**Proof for Lemma 3.1.** Let $P^\pi$ be the (stochastic) transition matrix for the MDP corresponding to a policy $\pi$, such that for any $x \in \mathbb{R}^{\mathcal{S}\times\mathcal{A}}$, $P^\pi x(s,a) = \mathbb{E}_{s'\sim\mathcal{P}(\cdot|s,a),a'\sim\pi(\cdot|s)}\left[x(s,a)\right]$.

Let $r = \mathcal{T}^\pi Q$. We expand $\mathcal{T}^\pi$ in vector form over $\mathcal{S}\times\mathcal{A}$ using $P^\pi$. Then $\boldsymbol{r} = \boldsymbol{Q} - \gamma P^\pi(\boldsymbol{Q} - \log\boldsymbol{\pi})$. Here, $(I-\gamma P^\pi)$ is invertible as $\|\gamma P^\pi\| < 1$, for $\gamma < 1$, and the corresponding Neumann series converges. Thus $\boldsymbol{Q} = (I-\gamma P^\pi)^{-1}(\boldsymbol{r} - \log\boldsymbol{\pi})$. So we see that for any $r$, there exists a unique image $Q$ proving that $\mathcal{T}^\pi$ is a bijection.

Furthermore, on rearranging the vector form, we have $\boldsymbol{Q} = \boldsymbol{r} + \gamma P^\pi(\boldsymbol{Q} - \log\boldsymbol{\pi})$. This is just the vector expansion of the soft-bellmann operator $\mathcal{B}_r^\pi$, which has a unique contraction $Q$ for a given $r$. Thus, $Q = (\mathcal{T}^\pi)^{-1}r = \mathcal{B}_r^\pi Q$ for any $r$.

**Lemma A.1.** *Define $\mathcal{T}^* : \mathbb{R}^{\mathcal{S}\times\mathcal{A}} \to \mathbb{R}^{\mathcal{S}\times\mathcal{A}}$ such that*

$$(\mathcal{T}^*Q)(s,a) = Q(s,a) - \gamma\mathbb{E}_{s'\sim\mathcal{P}(\cdot|s,a)}[\log\sum_{a'}\exp Q(s',a')]$$

*Then $\mathcal{T}^*$ is bijective $\implies$ For $r = \mathcal{T}^*Q$, we can freely transform between $Q$ and $r$.*

*Proof.* For $r = \mathcal{T}^*Q$, we have $Q(s,a) = r(s,a) + \gamma\mathbb{E}_{s'\sim\mathcal{P}(\cdot|s,a)}[\log\sum_{a'}\exp Q(s',a')]$. This is just the soft bellman equation, for which a unique contraction $Q^*$ exists satisfying it [14]. Thus for any $r$, we have a unique image $Q^*$ corresponding to it such that $r = \mathcal{T}^*Q^*$.

Hence, $\mathcal{T}^*$ is a bijection. □

**Lemma A.2.** *Let the initial state distribution be $p_0(s)$, then for a policy $\pi$ and $V^\pi$ defined as before, we have*

$$\mathbb{E}_{(s,a)\sim\rho}[V^\pi(s) - \gamma\mathbb{E}_{s'\sim\mathcal{P}(\cdot|s,a)}V^\pi(s')] = (1-\gamma)\mathbb{E}_{s\sim p_0}[V^\pi(s)]$$

*Proof.* We expand the discounted stationary distribution $\rho$ over state-actions and show the series forms a telescopic sum.

Let $p_t^\pi(s)$ be the marginal state distribution at time $t$ for a policy $\pi$.

Then,

$$\begin{aligned}
&\mathbb{E}_{(s,a)\sim\rho}[V^\pi(s) - \gamma\mathbb{E}_{s'\sim\mathcal{P}(\cdot|s,a)}V^\pi(s')]\\
&= (1-\gamma)\textstyle\sum_{t=0}^\infty\gamma^t\mathbb{E}_{s\sim p_t^\pi,a\sim\pi(s)}\left[V^\pi(s) - \gamma\mathbb{E}_{s'\sim\mathcal{P}(\cdot|s,a)}V^\pi(s')\right]\\
&= (1-\gamma)\textstyle\sum_{t=0}^\infty\gamma^t\mathbb{E}_{s\sim p_t^\pi}[V^\pi(s)] - (1-\gamma)\sum_{t=0}^\infty\gamma^{t+1}\mathbb{E}_{s\sim p_{t+1}^\pi}[V^\pi(s)]\\
&= (1-\gamma)\mathbb{E}_{s\sim p_0}[V^\pi(s)]
\end{aligned}$$

□

**Corollary A.2.1.** *In fact, for any valid occupancy measure $\mu$ over state-actions and $V^\pi$, it holds that*

$$\mathbb{E}_{(s,a)\sim\mu}[V^\pi(s) - \gamma\mathbb{E}_{s'\sim\mathcal{P}(\cdot|s,a)}V^\pi(s')] = (1-\gamma)\mathbb{E}_{s\sim p_0}[V^\pi(s)]$$

*Proof.* This relies on the fact that $V^\pi(s)$ is a function of only state and doesn't depend on the action. First, for any valid occupancy measure $\mu$, there exists a corresponding unique policy $\beta^\mu(a|s)$ s.t. $\beta^\mu$ generates $\mu$ [17].

Let $p_t^\mu(s)$ be the marginal state distribution at timestep $t$ for the policy $\beta^\mu$. Then,

$$\begin{aligned}
&\mathbb{E}_{(s,a)\sim\mu}[V^\pi(s) - \gamma\mathbb{E}_{s'\sim\mathcal{P}(\cdot|s,a)}V^\pi(s')]\\
&= (1-\gamma)\textstyle\sum_{t=0}^\infty\gamma^t\mathbb{E}_{s\sim p_t^\mu,a\sim\beta^\mu(s)}\left[V^\pi(s) - \gamma\mathbb{E}_{s'\sim\mathcal{P}(\cdot|s,a)}V^\pi(s')\right]\\
&= (1-\gamma)\textstyle\sum_{t=0}^\infty\gamma^t\mathbb{E}_{s\sim p_t^\mu}[V^\pi(s)] - (1-\gamma)\sum_{t=0}^\infty\gamma^{t+1}\mathbb{E}_{s\sim p_{t+1}^\mu}[V^\pi(s')]\\
&= (1-\gamma)\mathbb{E}_{s\sim p_0^\mu}[V^\pi(s)]
\end{aligned}$$

Now $p_0^\mu$ is just the initial state distribution $p_0$ which is independent of the policy, thus giving our result.

□

**Lemma A.3.** $\mathbb{E}_\rho[(\mathcal{T}^\pi Q)(s,a)] + H(\pi) = (1-\gamma)\mathbb{E}_{p_0}[V^\pi(s_0)]$*, where $p_0(s)$ is the initial state distribution.*

*Proof.* We can show this forms a telescopic series as in [28] using lemma A.2 to depend only on the initial state distribution:

$$\begin{aligned}\mathbb{E}_{\rho}[Q(s,a)-\gamma\mathbb{E}_{s'\sim\mathcal{P}(\cdot|s,a)}V^{\pi}(s')]+H(\pi) &= \mathbb{E}_{\rho}[Q(s,a)-\gamma\mathbb{E}_{s'\sim\mathcal{P}(\cdot|s,a)}V^{\pi}(s')+H(\pi(a|s))]\\ &= \mathbb{E}_{\rho}[Q(s,a)-\log\pi(a|s)-\gamma\mathbb{E}_{s'\sim\mathcal{P}(\cdot|s,a)}V^{\pi}(s')]\\ &= \mathbb{E}_{\rho}[V^{\pi}(s)-\gamma\mathbb{E}_{s'\sim\mathcal{P}(\cdot|s,a)}V^{\pi}(s')]\\ &= (1-\gamma)\mathbb{E}_{p_0}[V^{\pi}(s)],\end{aligned}$$

This makes sense as the LHS and RHS both represent the max entropy RL objective, that is to maximize the cumulative sum of rewards or the expected value with respect to a policy for the initial state. ☐

**Lemma A.4.** *SAC actor update decreases the objective $\mathcal{J}(\pi,Q)$ for the actor-critic update in main paper, wrt $\pi$ for a fixed $Q$.*

*Proof.*

$$V^{\pi}(s)=\mathbb{E}_{a\sim\pi}[Q(s,a)-\log\pi(a|s)]=-D_{KL}\left(\pi(\cdot|s)\|\frac{1}{Z_s}\exp(Q(s,\cdot)\right)+\log(Z_s),$$

where $Z_s$ is the normalizing factor $\sum_a \exp Q(s,a)$

Now, for a policy $\pi'$ the the SAC actor update rule [13] is $\arg\min_{\pi'} D_{KL}\left(\pi'\|\frac{1}{Z}\exp(Q)\right)$

Thus, if $\pi$ is the policy obtained on applying the SAC actor update to $\pi'$, we have $V^{\pi}(s)>V^{\pi'}(s)$. So, as long as $\phi$ in $\mathcal{J}$ is a monotonically non-decreasing function, this implies $\mathcal{J}(\pi,Q)<\mathcal{J}(\pi',Q)$. ☐

# B Appendix B

**Integral Probability Metric (IPM)** An IPM parameterized by $\mathcal{F}$ between two distributions $P$ and $Q$ is defined as

$$\gamma_{\mathcal{F}}(P,Q):=\sup_{f\in\mathcal{F}}|\mathbb{E}_P f(X)-\mathbb{E}_Q f(X)| \tag{17}$$

Suppose $\mathcal{F}$ is such that $f\in\mathcal{F}\Rightarrow -f\in\mathcal{F}$. Then,

$$\gamma_{\mathcal{F}}(P,Q)=\sup_{f\in\mathcal{F}}|\mathbb{E}_P f-\mathbb{E}_Q f|=\sup_{f\in\mathcal{F}}\ \mathbb{E}_P f-\mathbb{E}_Q f \tag{18}$$

Some IPMs that satisfy this symmetry are: Dudley metric, Wasserstein metric, total variation distance, Maximum Mean Discrepancy (MMD).

We can see that for $\phi=\mathcal{I}$, $R_\psi=\mathcal{F}$, Eq. 8 reduces to Eq. 18.

**$f$-divergence** The $f$-divergence between two distributions $P$ and $Q$ is defined using the convex conjugate $f^*$ as

$$D_f(P\|Q)=\mathbb{E}_Q\left[f\left(\frac{P}{Q}\right)\right]=\sup_{g:\mathcal{X}\to\mathbb{R}}\mathbb{E}_P[g(X)]-\mathbb{E}_Q\left[f^*(g(X))\right] \tag{19}$$

Interpreting $g=-r$

$$\begin{aligned}D_f(P\|Q) &= \sup_{r:\mathcal{X}\to\mathbb{R}}\ \mathbb{E}_P[-r(X)]-\mathbb{E}_Q\left[f^*(-r(X))\right] &(20)\\ &= \sup_{r:\mathcal{X}\to\mathbb{R}}\ \mathbb{E}_Q\left[-f^*(-r)\right]-\mathbb{E}_P[r] &(21)\end{aligned}$$

Thus, for $\phi(x)=-f^*(-x)$, $R_\psi=\mathbb{R}^{\mathcal{S}\times\mathcal{A}}$ , Eq. 8 reduces to Eq. 20.

Table 4: List of divergence functions, convex conjugates, $\phi$ and optimal reward estimators

| Divergence | $f(t)$ | $f^*(u)$ | $\phi(x)$ | $r$ |
|---|---|---|---|---|
| Forward KL | $-\log t$ | $-1-\log(-u)$ | $1+\log x$ | $\frac{\rho_E}{\rho}$ |
| Reverse KL | $t\log t$ | $e^{(u-1)}$ | $-e^{-(x+1)}$ | $-(1+\log\frac{\rho}{\rho_E})$ |
| Squared Hellinger | $(\sqrt{t}-1)^2$ | $\frac{u}{1-u}$ | $\frac{x}{1+x}$ | $\sqrt{\frac{\rho_E}{\rho}}-1$ |
| Pearson $\chi^2$ | $(t-1)^2$ | $u+\frac{u^2}{4}$ | $x-\frac{x^2}{4}$ | $2(1-\frac{\rho}{\rho_E})$ |
| Total variation | $\frac{1}{2}\|t-1\|$ | $u$ | $x$ | $\frac{1}{2}\,\mathrm{sign}\,(1-\frac{\rho}{\rho_E})$ |
| Jensen-Shannon | $-(t+1)\log(\frac{t+1}{2})+t\log t$ | $-\log(2-e^u)$ | $\log(2-e^{-x})$ | $\log\frac{1}{2}(1+\frac{\rho_E}{\rho})$ |

### B.1 Implementation of Statistical Distances

**Total Variation** Total variation gives a constraint on reward functions: $|r| \le \frac{1}{2}$.

As $Q_{t'} = \sum_{t=t'}^{\infty} \gamma^t r(s_t, a_t) + \gamma^t H(a_t|s_t)$, we obtain a constraint on $Q$:

$|Q| \le \frac{1}{1-\gamma}(R_{max} + \log|A|) = \frac{1}{1-\gamma}(\frac{1}{2} + \log|A|)$

This can be easily enforced by bounding $Q$ to this range using a $tanh$ activation.

**$W_1$ Distance** For Wasserstein-1 distance, we use gradient penalty [12] to enforce the Lipschitz constraint, although other techniques like spectral normalization [26] can also be utilized.

**$\chi^2$-divergence** $\chi^2$-divergence corresponds to a $f$-divergence with a choice of $f(x) = (x-1)^2$.

We generalize this to a choice of $f(x) = \alpha(x-1)^2$ with $\alpha > 0$, which scales the original divergence by a constant factor of $\alpha$.

Then $\phi(x) = -f^*(-x) = x - \frac{1}{4\alpha}x^2$. It corresponds to using a (strong) convex reward regularizer $\psi(r) = \frac{1}{4\alpha}r^2$.

### B.2 Effect of different Divergences

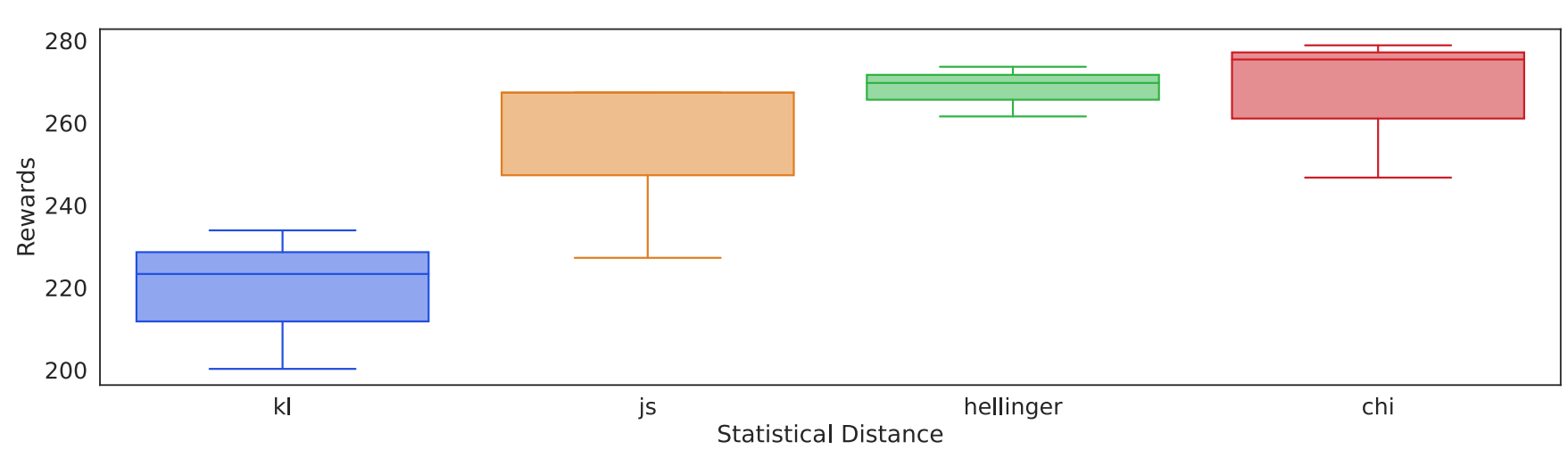

Figure 6: **Divergence ablation.** We show environment returns for different divergences on LunarLander.

We test IQ-Learn with different divergences: Jensen-Shannon (JS), Hellinger, KL and $\chi^2$ divergence. We use the LunarLander environment with our offline IL experimental settings and a single expert trajectory. All experiments are repeated over 10 seeds. We show a box-plot of the environment returns for different divergences and find that JS, Hellinger and $\chi^2$ divergence perform similarly, consistent with the findings on different type of GANs [25]. Here, KL-divergence performs worse and is suboptimal compared to the other divergences.

## C Appendix C

In this section, we expand over our analysis in Section 3 and present proof of properties over the $Q$-policy space: Propositions 3.3, 3.4, 3.5 in main paper.

For simplicity, we define a concave function $\phi : \mathbb{R} \to \mathbb{R} \cup \{-\infty\}$, such that $g$ is given as $g(x) := x - \phi(x)$. We are interested in regularizers $\psi$ induced by $g$, such that

$$\psi_g(r) = \mathbb{E}_{\rho_E}[g(r(s,a))] \tag{22}$$

We simplify the IRL objective (from Eq. 5):

$$\mathcal{J}(\pi, Q) = \mathbb{E}_{\rho_E}[\phi(Q - \gamma \mathbb{E}_{s' \sim \mathcal{P}(\cdot|s,a)} V^\pi(s'))] - (1-\gamma)\mathbb{E}_{\rho_0}[V^\pi(s_0)],$$

**Lemma C.1.** *$\mathcal{J}(\pi, \cdot)$ is concave for all $\pi \in \Pi$.*

*Proof.* Let $Q_1, Q_2 \in \Omega$ and suppose $\lambda \in [0,1]$. We rely on the fact that the regularized IRL objective $L(\pi, \cdot)$ is concave for all $\pi$. Note that $r = \mathcal{T}^\pi Q$ is an affine transform of $Q$, given in vector form as $\boldsymbol{r} - \log \boldsymbol{\pi} = (I - P^\pi)\, \boldsymbol{Q}$. Thus, $\mathcal{T}^\pi(\lambda Q_1 + (1-\lambda)Q_2) = \lambda \mathcal{T}^\pi Q_1 + (1-\lambda)\mathcal{T}^\pi Q_2$.

$$\begin{aligned}
\mathcal{J}(\pi, \lambda Q_1 + (1-\lambda)Q_2) &= L(\pi, \mathcal{T}^\pi(\lambda Q_1 + (1-\lambda)Q_2)) \\
&= L(\pi, \lambda \mathcal{T}^\pi Q_1 + (1-\lambda)\mathcal{T}^\pi Q_2) \\
&\geq \lambda L(\pi, \mathcal{T}^\pi Q_1) + (1-\lambda) L(\pi, \mathcal{T}^\pi Q_2) \\
&= \lambda \mathcal{J}(\pi, Q_1) + (1-\lambda)\mathcal{J}(\pi, Q_2)
\end{aligned}$$

Thus, $\mathcal{J}(\pi, \cdot)$ is concave. □

**Lemma C.2.** *For $\psi_g$ corresponding to a non-decreasing $\phi$, $\mathcal{J}(\cdot, Q)$ is quasiconvex for all $Q \in \Omega$, and has a unique minima $\pi_Q = \frac{1}{Z_s}\exp(Q)$ with normalizing factor $Z_s = \sum_a \exp Q(s,a)$.*

*Proof.* We have,

$$V^\pi(s) = \mathbb{E}_{a\sim\pi}[Q(s,a) - \log \pi(a|s)] = -D_{KL}\left(\pi(\cdot|s) \| \frac{1}{Z_s}\exp(Q(s,\cdot)\right) + \log(Z_s),$$

For a fixed $Q$, the KL divergence is strictly convex in $\pi$ with minima at $\pi_Q$, implying $V^\pi(s)$ is strictly concave in $\pi$ . Similarly, $r = \mathcal{T}^\pi Q = Q - \gamma \mathbb{E}_{s'\sim\mathcal{P}(\cdot|s,a)} V^\pi(s')$ is strictly convex in $\pi$ with minima at $\pi_Q$. Now, $\mathbb{E}_{\rho_E}[\phi(\mathcal{T}^\pi Q)]$ will be minimum at $\pi_Q$ and will be always non-decreasing as we pull away. Thus $\mathcal{J}(\pi, Q) > \mathcal{J}(\pi_Q, Q)$, for any $\pi \neq \pi_Q$. This is sufficient to establish the quasiconvexity of $J(\cdot, Q)$ with a unique minima at $\pi_Q$. □

Now $\Pi$ is compact and convex and $\mathbb{R}^{\mathcal{S}\times\mathcal{A}}$ is convex. As $\mathcal{J}(\pi, .)$ is concave, it is also quasiconcave for all $\pi$, and $\mathcal{J}(\cdot, Q)$ is quasiconvex for all $Q$. Thus, we can use Sion's minimax theorem [36]:

$$\min_{\pi\in\Pi} \max_{Q\in\Omega} \mathcal{J}(\pi, Q) = \max_{Q\in\Omega} \min_{\pi\in\Pi} \mathcal{J}(\pi, Q),$$

implying the existence of a saddle point for $\mathcal{J}$.

Let $(\pi^*, r^*)$ be the saddle point for $L$. Then, as $r = \mathcal{T}^\pi Q$ is an affine transform of $Q$ for a fixed $\pi$, we have

$$\pi^* = \operatorname*{argmin}_{\pi\in\Pi} \max_{r\in\mathcal{R}} L(\pi, r) = \operatorname*{argmin}_{\pi\in\Pi} \max_{Q\in\Omega} L(\pi, \mathcal{T}^\pi Q) = \operatorname*{argmin}_{\pi\in\Pi} \max_{Q\in\Omega} \mathcal{J}(\pi, Q)$$

Thus, the first coordinate of the saddle points for $L$ and $\mathcal{J}$ coincides. Now, we can relate the second coordinates, using the affine transformation property:

$$r^* = \operatorname*{argmax}_{r\in\mathcal{R}} L(\pi^*, r) = \mathcal{T}^{\pi^*}\left(\operatorname*{argmax}_{Q\in\Omega} L(\pi^*, \mathcal{T}^\pi Q)\right) = \mathcal{T}^{\pi^*}\left(\operatorname*{argmax}_{Q\in\Omega} \mathcal{J}(\pi^*, Q)\right) = \mathcal{T}^{\pi^*} Q^*$$

Therefore, the saddle point of $L$ uniquely corresponds to the saddle point $(\pi^*, Q^*)$ of $\mathcal{J}$, given as $(\pi^*, r^*)$ for $r^* = \mathcal{T}^{\pi^*} Q^*$.

This forms the proof for Proposition 3.3, 3.4.

**Proof for Proposition 3.5** We have,

$$\mathcal{J}^*(Q) = \mathbb{E}_{\rho_E}[\phi(Q(s,a) - \gamma \mathbb{E}_{s' \sim \mathcal{P}(\cdot|s,a)} V^*(s'))] - (1-\gamma)\mathbb{E}_{p_0}[V^*(s_0)],$$

As log-sum-exp is convex, $V^*(s) = \log \sum_a \exp Q(s,a)$ is convex in Q. Then concavity follows from the fact that for the first term, $\phi(Q(s,a) - \gamma \mathbb{E}_{s' \sim \mathcal{P}(\cdot|s,a)} V^*(s'))$ is concave, as it is a concave function composed with a non-decreasing concave function.

### C.1 Generalization

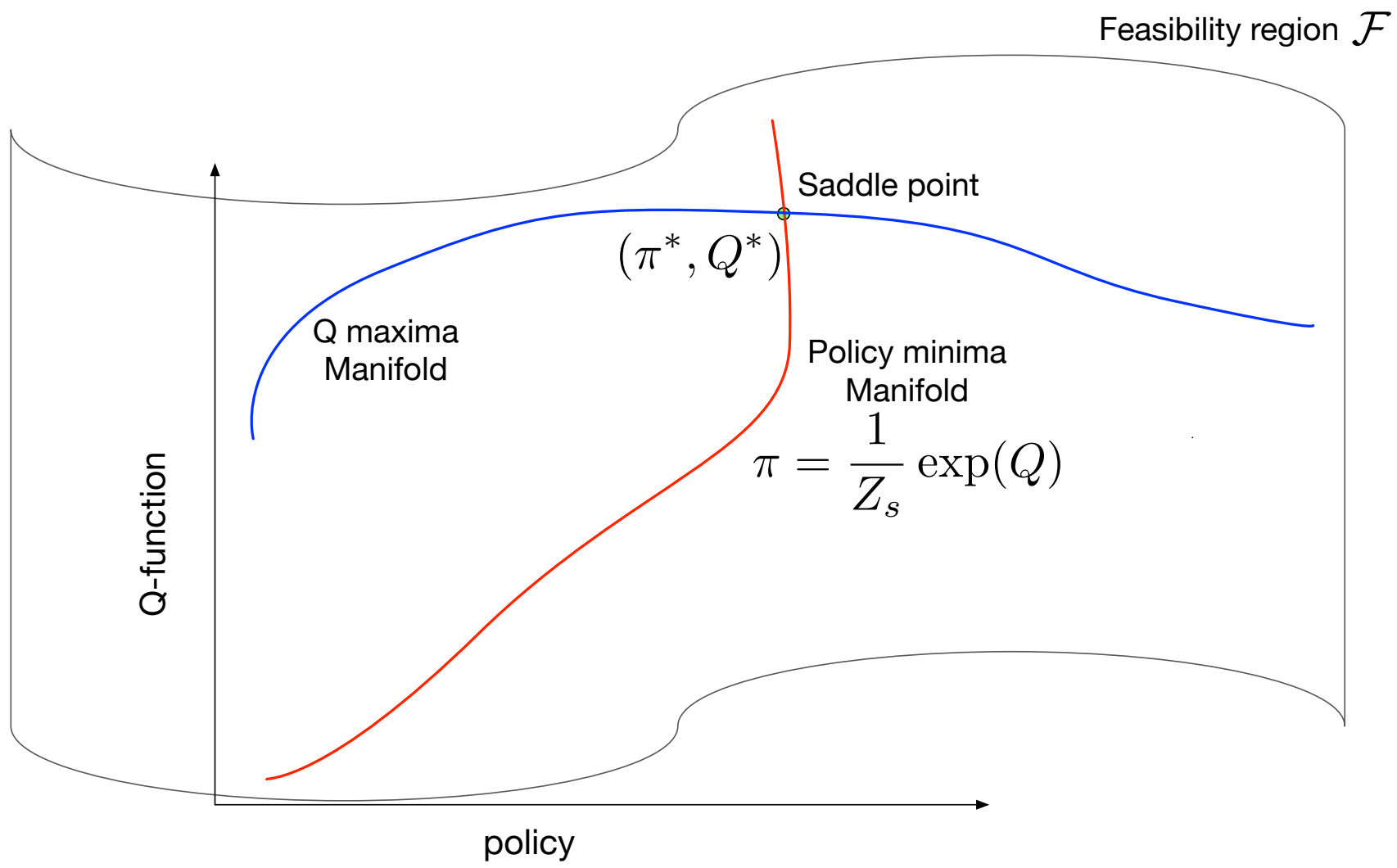

Figure 7: **Feasibility region in Q-policy space.**

In the above section, we made a monotonicity assumption on $\phi$ in Lemma C.2. We show that we can relax this assumption and the saddle point properties still hold, although $\mathcal{J}$ is not so well-behaved everywhere anymore.

For a fixed $\pi$, the optimizer of the concave problem, $\max_r L(\pi, r) = \mathbb{E}_{\rho_E}[\phi(r(s,a))] - \mathbb{E}_{\rho}[r(s,a)] - H(\pi)$ satisfies [9]:

$$\phi'(r)\rho_E - \rho = 0$$

Thus, $\phi'(r(s,a)) = \rho(s,a)/\rho_E(s,a) \in [0, \infty)$. This tells us that there exists a set of rewards $\mathcal{R}_\phi$, such that $\phi$ is non-decreasing on this set. For a concave $\phi$, $\mathcal{R}_\phi$ is just be the convex set of reals that are on the left of its maxima.

**Lemma C.3.** *Define a convex **feasibility region** on the Q-policy space:*

$$\mathcal{F}_\phi = \{(\pi, Q) : r = \mathcal{T}^\pi Q \in \mathcal{S} \times \mathcal{A} \to \mathcal{R}_\phi\}$$

*then for a given $\pi$, any optimal $Q = \text{argmax}_{Q'} \mathcal{J}(\pi, Q')$ has to lie in $\mathcal{F}_\phi$.*

*Proof.* If $Q$ is optimal, then $\mathcal{T}^\pi Q$ maximizes $L(\cdot, r)$, and so it's corresponding $r$ is optimal. For a fixed $\pi$, and any $(s,a) \in \mathcal{S} \times \mathcal{A}$, the optimal reward has to satisfy $\phi'(r(s,a)) = \rho(s,a)/\rho_E(s,a)$. Thus, each component of the reward vector lies in $\mathcal{R}_\phi$. This tells us $\mathcal{T}^\pi Q$ lies in the required region. □

We get two properties in the feasibility region $\mathcal{F}_\phi$:

1. $\text{argmax}_Q \mathcal{J}(\cdot, Q)$ lies in $\mathcal{F}_\phi$

[9] A concave function may not be differentiable everywhere and in general, we get a condition on the subdifferential of $\phi$: $\rho/\rho_E \in \partial\phi(r)$

2. $\phi$ is non-decreasing, so lemma C.2 holds in this region

We just need one last lemma to prove the existence of a unique saddle point:

**Lemma C.4.** *A saddle point exists only at the intersection of two curves:* $\operatorname{argmax}_Q \mathcal{J}(\cdot, Q)$ *and* $\operatorname{argmin}_\pi \mathcal{J}(\pi, \cdot)$

*Proof.* We parameterize the curves $f(\pi) = \operatorname{argmax}_Q \mathcal{J}(\pi, Q)$ and $g(Q) = \operatorname{argmin}_\pi \mathcal{J}(\pi, Q)$. A saddle point has to satisfy $\min_\pi \max_Q \mathcal{J}(\pi, Q) = \max_Q \min_\pi \mathcal{J}(\pi, Q)$. This implies, $\min_\pi \mathcal{J}(\pi, f(\pi)) = \max_Q \mathcal{J}(g(Q), Q)$. This equation can only be satisfied when both the curves intersect.

Therefore, any saddle point lies at the intersection of the Q-maxima and policy minima curves. □

We have established that within the feasibility region $\mathcal{F}_\phi$, lemma C.1 and C.2 hold. Thus, there exists a single saddle point in this region. Furthermore, $\operatorname{argmax}_Q \mathcal{J}(\cdot, Q)$ lies in $\mathcal{F}_\phi$ so lemma C.4 tells us there cannot exist any other saddle points outside $\mathcal{F}_\phi$.

This completes our proof of the existence of a unique saddle point of $\mathcal{J}$ for any concave $\phi$.

We summarize these properties in Fig 7.

### C.2 Convergence Guarantee

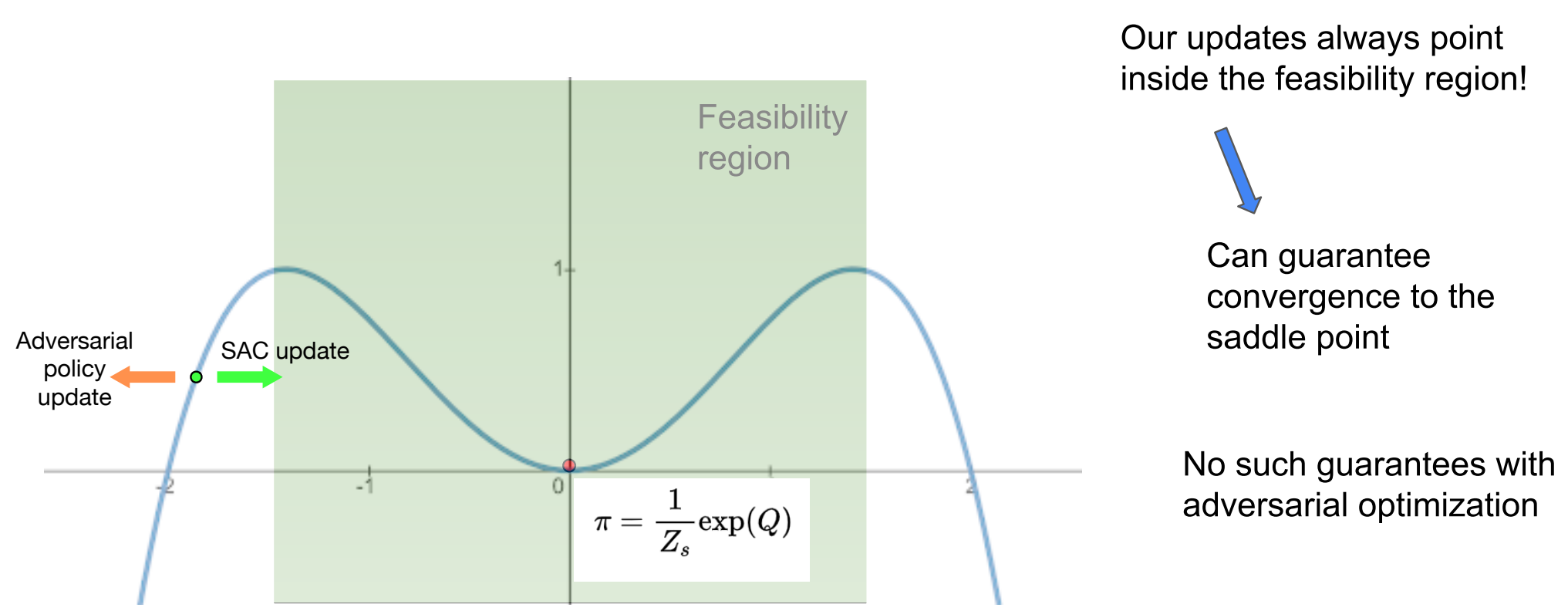

Figure 8: **Policy learning.** Comparison of SAC vs adverserial policy update outside the feasibility region.

For any $\phi$, our soft actor-critic (SAC) policy update (Sec 4.4) minimizes the KL divergence between the current policy $\pi$ and $\pi_Q$, always pointing towards the the policy minima manifold whereas adversarial policy update relying on the local gradient can diverge away from it (outside the feasibility region). This has the effect, that with sufficient steps, learning with SAC updates is guaranteed to converge to the saddle point, but no such guarantee exists with adversarial policy updates.

### C.3 Effect of various divergences

In the $Q$-policy space, the policy minima manifold $\pi_Q$ is an energy-based model of $Q$, and doesn't depend on the choice of regularizer $\psi$.

Whereas, the $Q$-maxima manifold is dependent on the choice of regularizer. As the saddle point is formed by the intersection of these two curves (Lemma C.4), we can study how different divergences will affect the saddle point which solves the regularized-IRL problem.

We have that for a choice of $\phi$, the $Q$-maxima manifold is given by the condition:

$$\phi'(r)\rho_E - \rho = 0$$

Thus on the maxima manifold, $r = (\phi')^{-1}(\rho/\rho_E)$. We visualize this in the Fig. 9, we see that different statistical distances correspond to different saddle points. The overall effect is that that at

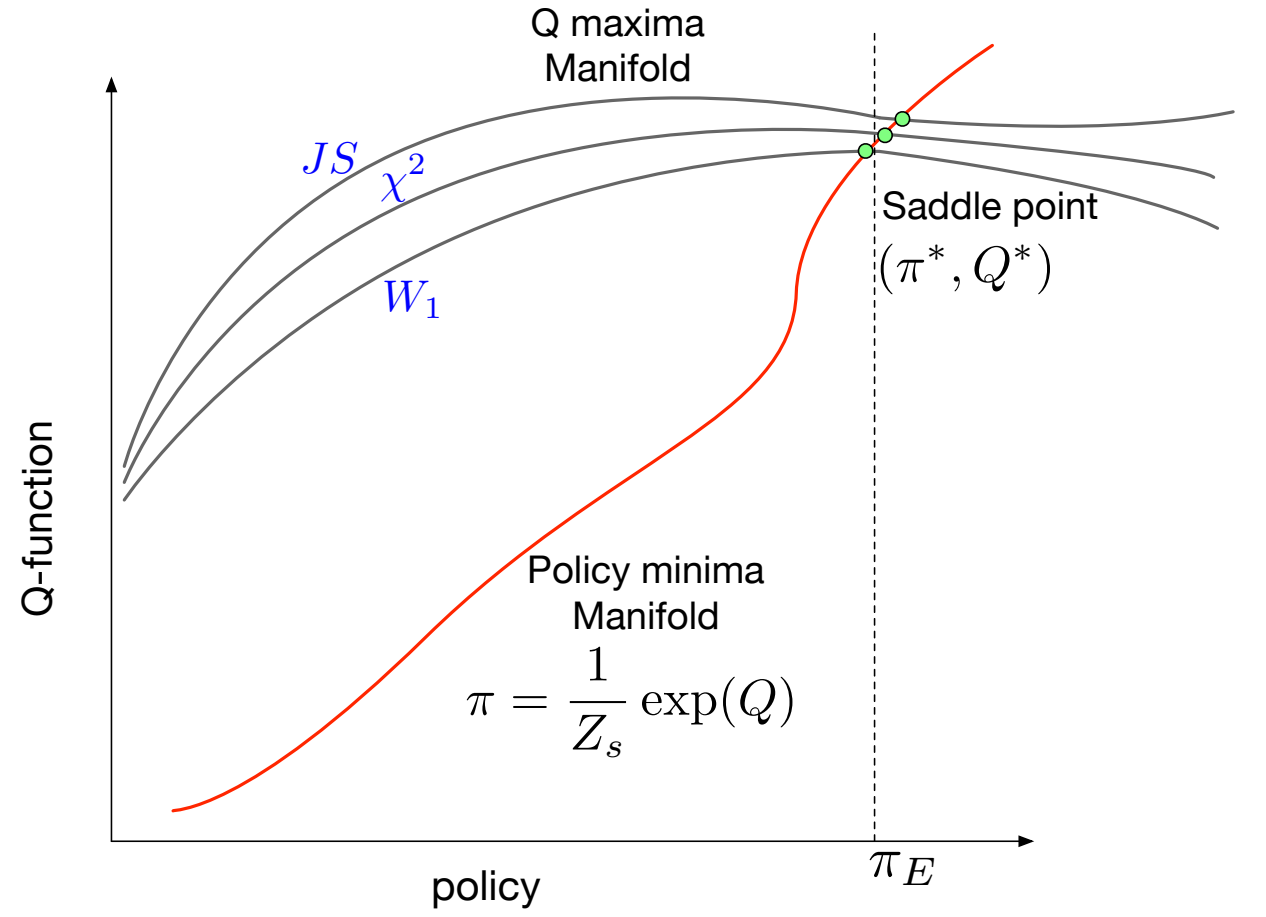

Figure 9: **Saddle points.** Effect of regularizer $\psi$ on the saddle point. (not to scale)

the saddle point $\pi^*$ remains close to $\pi_E$, but may not be exactly equal as the regularization constrains the policy class.

In general, $\pi^*$ is the solution to the (transcendental) equation:

$$\phi'(\mathcal{T}^\pi Q)\rho_E - \rho_Q = 0, \tag{23}$$

where $\rho_Q$ is the occupancy measure corresponding to $\pi_Q = \frac{1}{Z}\exp Q$.

For $f$-divergences, this can be simplified as

$$\mathcal{T}^\pi Q = -f'\left(\frac{\rho_Q}{\rho_E}\right) \tag{24}$$

For an IPM parametrized by $\mathcal{F}$, $\phi'(x) = 0$ and the equation will be maximized on the boundary of $\mathcal{F}$, without a closed form equation.

Now, SQIL [33] uses the reward of the form $1 - 0$ dependent on sampling from the expert or policy distributions. This condition corresponds to a maxima manifold in this space, such that instead of the reward being a function of the ratio density of the expert and the policy, it is stochastically dependent on the sampling. Thus, instead of being fixed, the manifold will shift stochastically with the sampling. This has the corresponding effect of shifting the saddle point and can result in numerical instabilities near convergence, as a unique convegence point does not exist for the SQIL style update.

Similary, we can analyze ValueDICE [22]. ValueDICE mimimizes the Reverse-KL divergence between the expert and policy using the Donsker-Varadhan (DV) variational form of Reverse-KL. This corresponds to the maxima manifold with rewards satisfying $r = \log(\rho_E/\rho)$, but suffers from two issues: 1) biased gradient estimates, and 2) adversarial policy updates.

We have already shown how adverserial policy updates are not optimal, we will now focus on fixing the biasing issue with the Reverse-KL distance.

First, the DV representation is given as:

$$KL(\rho, \rho_E) = \max_{r\in\mathcal{R}} \log \mathbb{E}_{\rho_E}[e^{-r(s,a)}] - \mathbb{E}_{\rho}[r(s,a)]$$

This corresponds to a $\phi(x) = \log \mathbb{E}_{\rho_E}[e^{-x}]$, even though its outside the class of $\psi$ we study, it satisfies all the previous properties we developed (Lemma C.1 - C.4).

Now, to unbias the Reverse-KL representation, we propose using the $f$-divergence representation, with $f(t) = t\log t - t + 1$. Then the $f$-divergence for this choice of $f$ is just the Reverse-KL divergence, but it's variational form is:

$$\max_{r\in\mathcal{R}} \mathbb{E}_{\rho_E}[-e^{-r(s,a)}] - \mathbb{E}_{\rho}[r(s,a)]$$

and corresponds to $\phi(x) = -e^{-x}$ with rewards $r = \log(\rho_E/\rho)$.

Thus, we can obtain the same $Q$-maxima manifold to minimize the Reverse-KL distance as ValueDICE by using this new representation, while avoiding the biasing issue.

**Effect of different forms of Reverse-KL** We test IQ-Learn with different variational representations of Reverse-KL: Donsker-Varadhan (DV), Original KL (KL), ours Modified KL (KL-fix). We use the LunarLander environment with our offline IL experimental settings and a single expert trajectory. All experiments are repeated over 10 seeds. We show a box-plot of the environment returns for different variational forms and find that our proposed form (KL-fix) and the DV representation perform similarly. The original f-divergence form of KL remains problematic, performing noticeably worse, which may be due to an issue with its corresponding Q-maxima manifold. Compared to DV, our proposed KL variation representation has the advantage of giving unbiased gradient estimates and can be more stable.

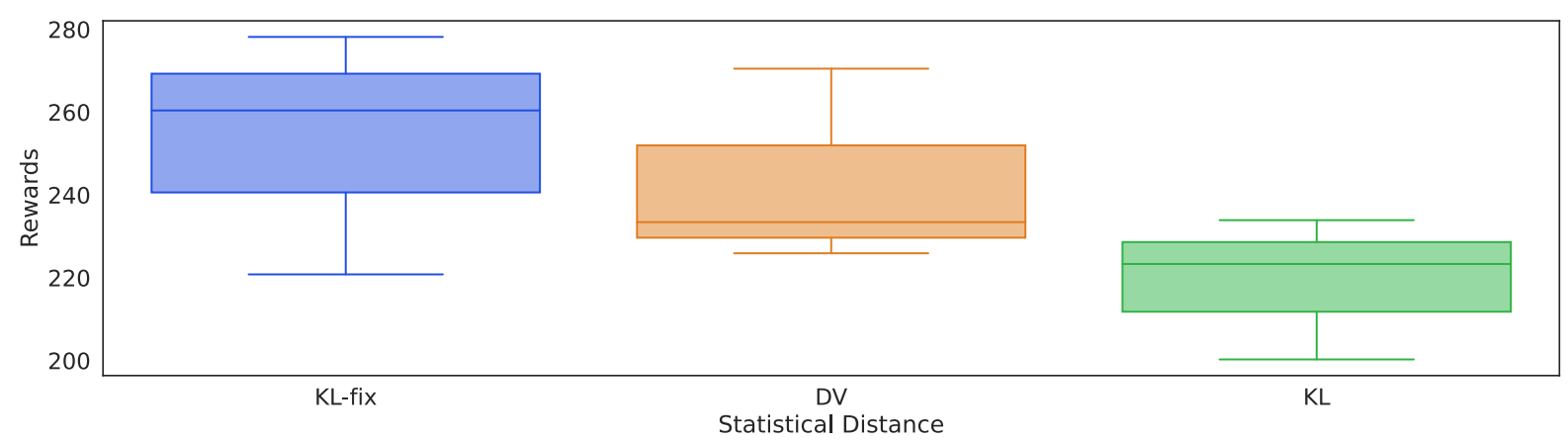

Figure 10: **Reverse-KL ablation.** We show environment returns for different variational forms of Reverse-KL on LunarLander.

# D Appendix D

## D.1 Implementation Details

For reproducibility, we plan to release all our expert demonstrations, either trained from scratch or obtained using Stable Baslines3 Zoo [32]. We also release an efficient expert data generation and data-loading pipeline, that can work with pre-trained Stable Baselines3 models, or arbitary pytorch RL agents. We hope this will make benchmarking for IL easier and help with standardization.

### D.1.1 Offline Setup

We mimic the settings used by prior works [19, 6] to make our results directly comparable for offline IL.

**Expert Demonstrations** We obtain expert demonstrations by training a DQN [27] agent from scratch for all the environments tested. Our trajectories were then sub-sampled for every 20th step in Acrobot and CartPole, and every 5th step in LunarLander.

**Training Setup** We test with (1,3,7,10,15) expert trajectories uniformly sampled from a pool of 1000 expert demonstrations. Each algorithm is trained until convergence and tested by performing 300 live rollouts in the simulated environment and recording the average episode rewards. We repeat this over 10 seeds, consequently with different initializations and seen trajectories.

**Implementation** All methods use neural networks with the same architecture of 2 hidden layers of 64 units each connected by exponential linear unit (ELU) activation functions.

We use the original public code implementations of EDM, AVRIL and ValueDICE. Note, ValueDICE is adapted to discrete environments using an actor with Gumbel-softmax distribution output.

**Hyperparameters** We use batch size 32 and $Q$-network learning rate $1e{-}4$ with entropy coefficient $0.01$. We found learning rate of $1e-4$ worked best for IQ-Learn on discrete environments. We also found entropy coefficient values $[1e-2, 1e-3]$ to be optimal depending on the environment. Here, we don't use target updates as we found them to give no visible improvement and slow down the training.

### D.1.2 Online Setup

**Expert Demonstrations** For Mujoco environments, we generate expert demonstrations from scratch using a Pytorch implementation of SAC. For Atari, we generate demonstrations using pre-trained DQN agents from Stable Baselines3 Zoo. For both, we generate a pool of 30 expert demonstrations and sample trajectories uniformly. For Mujoco results, we sample 1 expert demo and for Atari we sample 20 expert demos without any subsampling.

**Implementation** For Mujoco, with all methods we use critic and actor networks with an MLP architecture with 2 hidden layers and 256 hidden units, keeping settings similar to original SAC [13]. For Atari, with all methods we use a single convolution neural network same as the original DQN architecture [27]. For IQ-Learn in continuous environments, for SAC policy updates we sample states from both policy and expert distributions. We regularize policy states in addition to expert states to improve the stability of learning $Q$-values. We use soft target updates and find them helpful for stabilizing the training.

For BC and GAIL, we use the stable-baselines implementations. For SQIL, we use original public code for Atari environments. For ValueDICE, we use the open-sourced official code.

**Hyperparameters** For SAC style learning, we use default settings of critic learning rate $3e-4$ and policy learning rate values $[3e-4, 3e-5]$. We found $3e-5$ to work well in complex environments and remain stable, although $3e-4$ can be better with simpler environments (like Half-Cheetah). We use a fixed batch size of 256 and found entropy coefficient 0.01 to work well. We use soft target updates with the default SAC smoothing constant $\tau = 0.05$. For DQN-style learning on Atari, we use $Q$-network learning rate $1e-4$ with entropy coefficient $1e-4$ and batch size 64. We found entropy coefficient values $[1e-3, 1e-4]$ to work well. We didn't find noticeable improvements with using target updates on Atari (with the exception of Space Invaders, where they stabilize the training).

### D.2 Additional Results

**Mujoco** We show additional results on Mujoco obtained using 10 expert trajectories in Table 5. We find IQ-Learn gets state-of-art performance in all environments and reaches expert-level rewards.

Table 5: **Mujoco Results.** We show our performance on MuJoCo control tasks using 10 expert trajectories.

| Task | Random | BC | GAIL | ValueDICE | IQ (Ours) | Expert |
|---|---|---|---|---|---|---|
| Hopper | $14 \pm 8$ | $1345 \pm 422$ | $3322 \pm 510$ | $3399 \pm 651$ | $\mathbf{3529 \pm 15}$ | $3533 \pm 39$ |
| Half-Cheetah | $-282 \pm 80$ | $2701 \pm 950$ | $4280 \pm 1002$ | $4840 \pm 132$ | $\mathbf{5154 \pm 82}$ | $5098 \pm 62$ |
| Walker | $1 \pm 5$ | $3730 \pm 1440$ | $4417 \pm 420$ | $4384 \pm 345$ | $\mathbf{5212 \pm 85}$ | $5274 \pm 53$ |
| Ant | $-70 \pm 111$ | $2272 \pm 472$ | $3997 \pm 312$ | $4507 \pm 265$ | $\mathbf{4683 \pm 67}$ | $4700 \pm 80$ |
| Humanoid | $123 \pm 35$ | $2057 \pm 843$ | $372 \pm 51$ | $2001 \pm 524$ | $\mathbf{5288 \pm 73}$ | $5313 \pm 210$ |

**Atari Suite.** We show detailed performance of IQ-Learn on Atari Suite environments using 20 expert demonstrations in Table 6.

Table 6: **Results on Atari Suite**. We show our results on Atari Suite tasks using 20 expert demonstrations.

| Env | IQ (Ours) | Expert |
|---|---|---|
| Pong | $19 \pm 2$ | $21 \pm 0$ |
| Breakout | $320 \pm 72$ | $376 \pm 34$ |
| Space Invaders | $807 \pm 102$ | $823 \pm 272$ |
| BeamRider | $3025 \pm 845$ | $4295 \pm 1173$ |
| Seaquest | $2349 \pm 342$ | $2393 \pm 291$ |
| Qbert | $12940 \pm 2026$ | $11496 \pm 1988$ |

**Reward Correlations.** We show the Pearson correlation coefficient of our learnt rewards with environment rewards in Table 7.

Table 7: **Reward Correlations**. We show pearson correlations between our learnt reward and the env rewards.

| Env | Reward correlation |
|---|---|
| Cartpole | 0.99 |
| LunarLander | 0.92 |
| Hopper | 0.99 |
| Half-Cheetah | 0.86 |
| Pong | 0.67 |

**Do we overfit?** Compared to ValueDICE, we don't observe overfitting using IQ-Learn with the number of update steps. We show a comparision on Half-Cheetah environment using one expert trajectory in Fig 11. ValueDICE begins to overfit around 100k update steps, whereas IQ-Learn converges to expert rewards and remains stable.

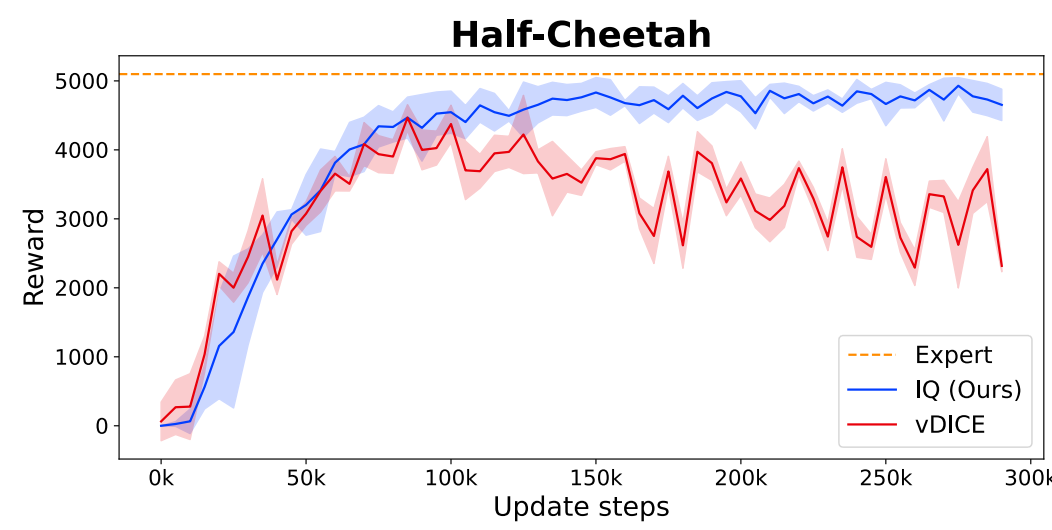

Figure 11: **Half-Cheetah overfitting comparision**

### D.3 Recovering Rewards

We show visualizations of our reward correlations on the Hopper environment using 10 expert demonstrations in Fig 12. We obtain a Pearson correlation of 0.99 of our recovered episode rewards compared with the original environment rewards, showing that our rewards are almost linear with the actual rewards, and thus can be used for Inverse RL. Note, that to recover rewards with IQ-Learn, we need to sample the current state and the next state.

We perform similar comparisons on GAIL and SQIL, obtaining Pearson coefficients of 0.90 and 0.72 respectively.

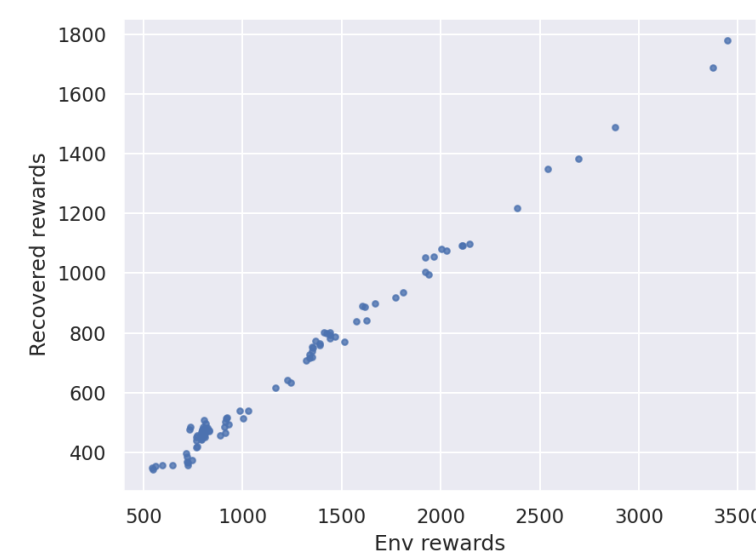

Figure 12: **Hopper correlations**

In the main paper, we also show recovered rewards on a simple grid environment by using sampling based $Q$-learning with a simple $Q$-network having two hidden layers. In the section below, we further compare IQ-Learn on a tabular setting.

**Tabular Inverse RL** To further validate IQ-Learn as a method for IRL and show we recover correct rewards, we directly compare with the classical Max Entropy IRL [43] method on a tabular Grid world setting, by using an open-source implementation[10]. We implement IQ-Learning as a modification to tabular value iteration. The classical method requires repeated backward and forward passes, to calculate soft-values and action probabilities for a given reward and optimize the rewards respectively. IQ-Learn skips the expensive backward pass and directly optimizes the rewards. We show comparision in Fig 13, where we find our method recovers very similar rewards while being more than 3x faster.

### D.4 Imitation learning with Observations

Table 8: **Results on ILO**. We show evironment returns using 1 and 10 expert demonstrations.

| Env | 1 demo | 10 demos |
|---|---|---|
| CartPole | $452 \pm 50$ | $485 \pm 25$ |
| LunarLander | $20 \pm 102$ | $220 \pm 69$ |
| Hopper | $2507 \pm 345$ | $3465 \pm 51$ |

We show results for IQ-Learn trained with using only expert observations in Table 8. We test on CartPole, LunarLander and Hopper environments with 1 and 10 expert demonstrations using online IL settings without any subsampling of trajectories. We find that with one expert demonstration, we get below expert-level rewards, and as expected, our performance suffers compared to with using expert actions. We find using 10 demonstrations is enough to reach expert-level performance in these simple environments.

Target updates are helpful in stabilizing the training in this setting.

[10] https://github.com/yrlu/irl-imitation

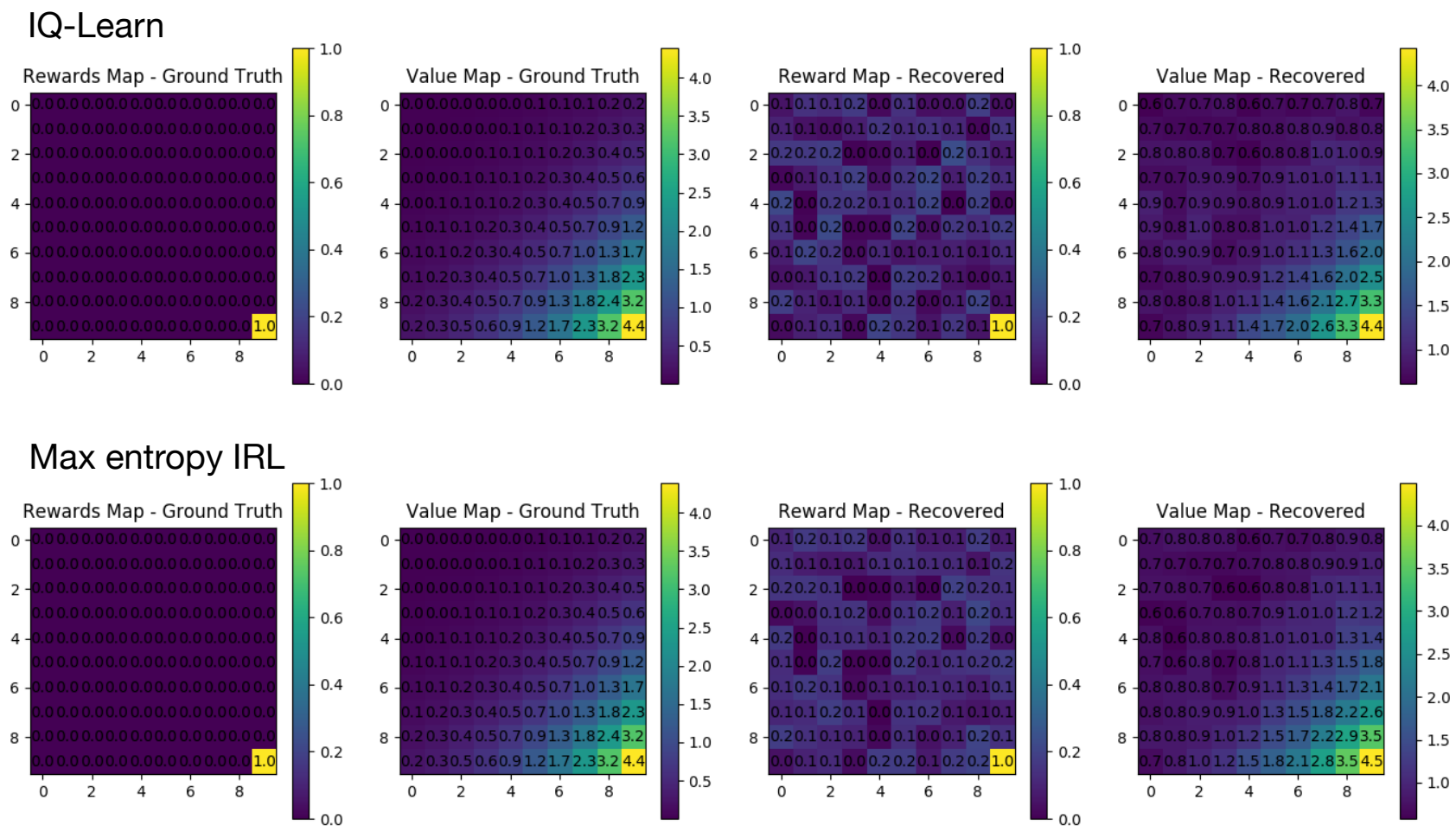

Figure 13: **Tabular Grid Rewards.** We recover similar rewards as Max entropy IRL (Ziebert et al.) while avoiding an expensive backward pass.

# E Appendix E

## E.1 Dynamics-Aware Imitation Learning and the Loop MDP

In this section we illustrate the importance of dynamics-awareness in imitation learning with a toy MDP based on the Loop MDP from [34]. The MDP is shown in Fig 14. The MDP has a fixed length of 100 steps. The key problem for dynamics-unaware algorithms, such as behavioural cloning, is the behaviour in state $s_2$. If we happen to use an expert trajectory where the expert never visits state $s_2$, then the learned policy will not necessarily have the right behaviour in state $s_2$. This is because the objective for behavioural cloning is to match the action probabilities in the expert states, and $s_2$ is not in the expert states visited. However, the dynamics-aware methods are able to deduce that taking action $a_1$ in state $s_2$ will return the imitator to state $s_1$. Although this MDP is simple, it illustrates a general advantage of dynamics-aware methods which will hold in many situations. In particular, it will hold for environments where the expert may keep very close to an optimal trajectory, yet it is possible to recover back to that trajectory if a small mistake is made, such as in autonomous lane-keeping in a car.

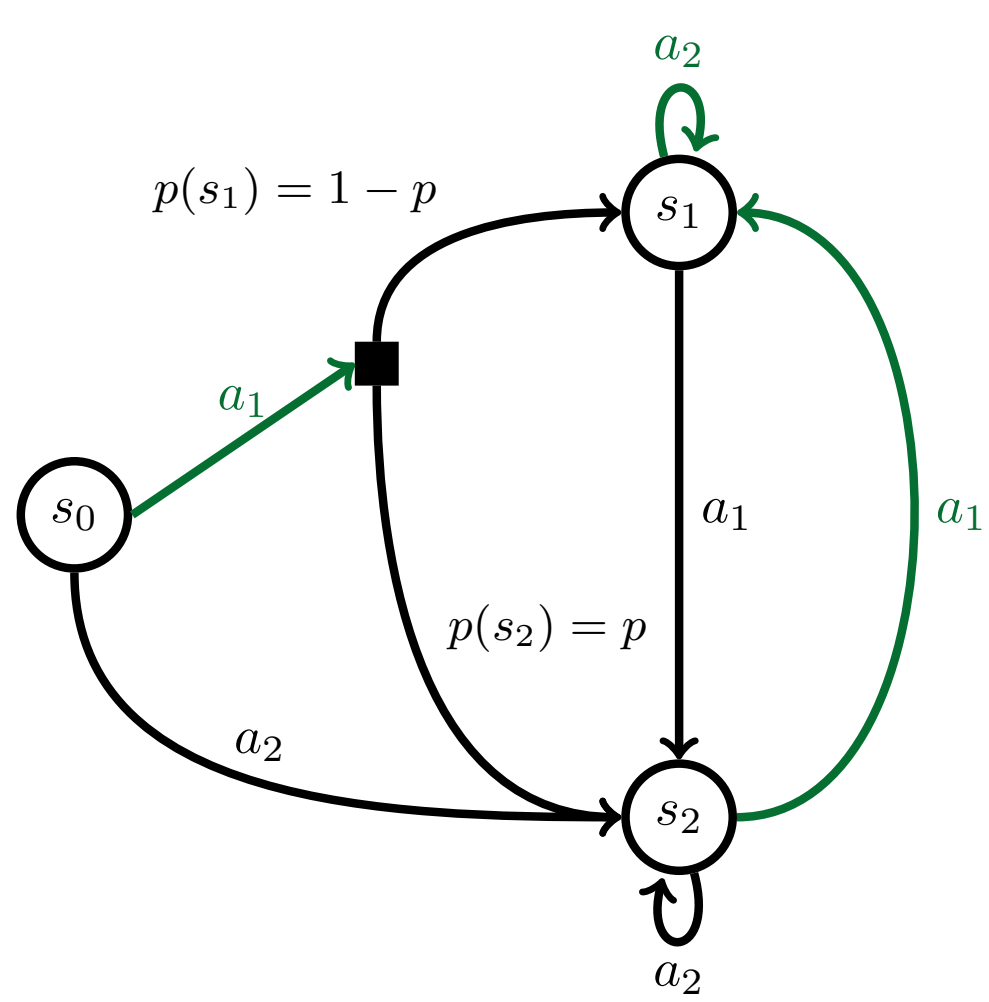

Figure 14: A variant of the Loop MDP from [34]. Taking actions labelled in green gives 1 reward, while actions in black give reward 0. The MDP is stochastic for action $a_1$ in state $s_0$, which with probability $p$ leads to state $s_2$, and with probability $1 - p$ leads to state $s_1$.

To substantiate this illustrative case, we implemented this MDP and evaluated a few methods. We use a single expert trajectory which goes from $s_0$ to $s_1$, never going to state $s_2$. We set $p = 0.5$ for this experiment. The results are in Table 9, averaged over five random seeds. They are as we expect, with the dynamics-aware methods able to convincingly master the environment and find the optimal policy, while the behavioural cloning approach achieves around 50 reward. This is because it learns the wrong behaviour in state $s_2$ so gets zero reward in that state in the 50% of the time that taking action $a_1$ results in a transition to state $s_2$.

Table 9: Results of imitation learning algorithms on the `Loop` MDP described above. We observe that the dynamics-unaware behavioural cloning baseline performs much worse than the other dynamics-aware methods.

| Method | Episode Reward |
|---|---|
| Behavioural Cloning | $54 \pm 5$ |
| SQIL | $100 \pm 0$ |
| IQ (Online, $\chi^2$) | $100 \pm 0$ |

### E.2 Ablation on Gamma

The dynamics are encoded in our learning objective by the discount factor $\gamma$, and setting it to zero removes dynamic-awareness in IQ-Learn.

To show how dynamics help with learning, we do an ablation on $\gamma$ with IQ-Learn. We use the offline IL settings for CartPole environment with one expert trajectory.

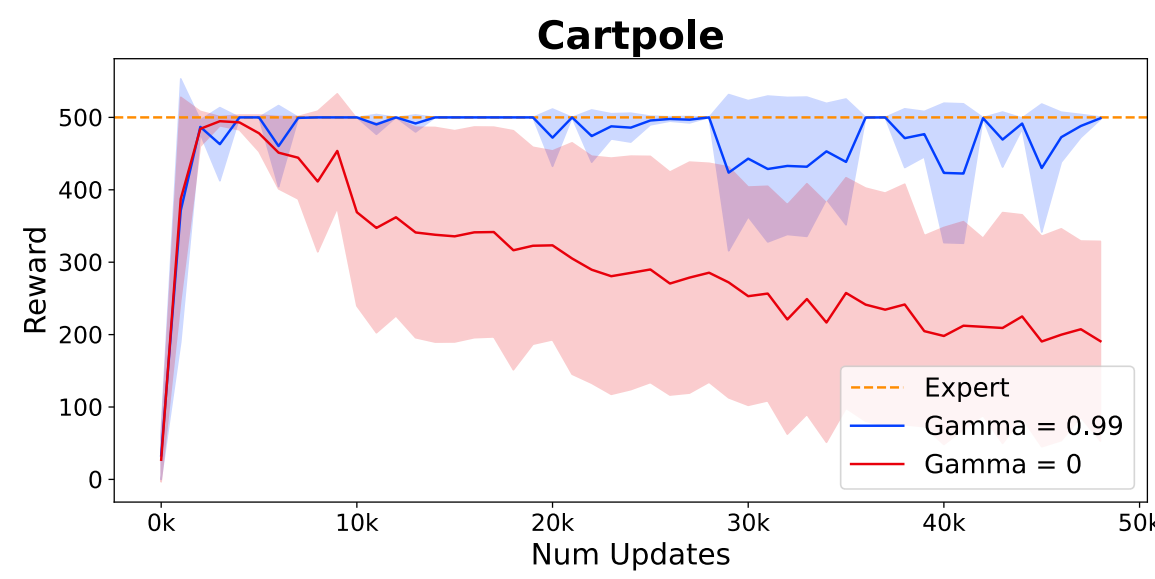

Figure 15: **Ablation on Gamma**

We set $\gamma$ to $0.99$ and $0$. The results are visualized in Fig 15, we can see that without the dynamics the training is not stable and there is a strong decay in the rewards obtained by the IL agent from the environment. Whereas, when using dynamics, we see that the training is stable and properly converges.

## F Appendix F

### F.1 Generalization over distribution shift

We show our method can be robust to distribution shifts between the expert and policy and perform additional experiments over two different settings: 1) Initial distribution shift using a modified LunarLander env motivated by [33] and 2) Goal distribution shift using DeepMind Control Suite.

#### F.1.1 Initial distribution shift

We experiment with initial shift distribution in the LunarLander-v2 environment similar to [33]. The agent is typically initialized in a small zone at the middle top of the screen. Instead, we modify the environment to initialize the agent near the top-left corner of the screen. We use experts from the unmodified environment, and test whether the agent can still learn to land the lunar lander while recovering from the initial distribution shift.

**Offline Case**: We find in the offline case that the agent cannot learn to recover from the occupancy shift. The lander typically tends to fly off the frame and shows random behavior. This is expected as IQ-learn is not aware of the shift of initial distributions between the agent and the expert, and can't explore the environment to correct the initial state shift to match the occupancy distributions.

**Online Case**: In the online case, we find that the agent can sufficiently explore the environment, and learns a behavior of first horizontally moving the lander from the top left to the top center and then successfully imitating the original expert trajectory, receiving an avg. episode reward of  250 with 10 expert demos.

An extra consideration here is in Eq. 9, where we originally only apply reward regularization to the expert states, but we find applying regularization to both expert and policy states to be beneficial in this case. As it enforces the learning of an implicit reward function that can generalize outside the expert distribution to more arbitrary policy states.

#### F.1.2 Goal Distribution Shift

We experiment with the *reacher_easy* task in DeepMind Control Suite. We choose the reacher environment as it is a multi-task environment, where the goal given by the target position changes in every episode randomly. Such environments have been found to be very difficult to solve using IRL [40] as a large number of expert demos are needed to fully cover the goal distributions, and usually require meta-IRL methods to figure the right task context for a given expert demonstration like PEMIRL [40].

We test with different number of expert demonstrations: $(1, 5, 10, 20)$ each with different target positions on the offline and online settings. The average expert performance is $\sim 990$ in this case and we report averaged results over 100 episodes with different targets.

**Offline Case**: In the offline setting, a single demonstration is typically not enough to learn a generalized reward function and leads to a reward that overfits to a particular target position. We quantify the results in Table 10, with the observation that imitation learning performance improves with the number of expert demos. This can be justified, as more experts with different targets allow learning a reward function that is better generalizable.

Table 10: **Offline**. We show evironment returns vs number of experts on *reacher_easy* for offline case.

| Num Experts | Rewards |
|---|---|
| 1 | 105.4 |
| 5 | 120.1 |
| 10 | 210.6 |
| 20 | 325.0 |

**Online Case**: In the online setting, our method is able to explore the environment over different episodes and can learn to correct the behavior leading to better performance. In particular, given a sufficient number of expert demos, it can learn to associate what expert behavior to imitate given a particular target and learns a more reward function generalizable over multiple goals. We show quantitative results in Table 11.

Table 11: **Online**. We show evironment returns vs number of experts on *reacher_easy* for online case.

| Num Experts | Rewards |
|---|---|
| 1 | 271.3 |
| 5 | 485.1 |
| 10 | 545.0 |
| 20 | 734.9 |
| 50 | 926.1 |

BC and GAIL on *reacher_easy* even with $50$ experts obtain mean rewards of $325.2$ and $440.1$ respectively, which is equivalent to what we see using our method with just $5$ expert demos! It is surprising to us that our method can learn a reward to figure out what goal state to reach, acting as a **meta-learner** even when not engineered specifically to do so.