# OpenReview forum: "IQ-Learn: Inverse soft-Q Learning for Imitation"
_NeurIPS.cc/2021/Conference — NeurIPS 2021 Spotlight_

### Official Review · Reviewer_rfsf · 2021-07-12

**Rating:** 8
**Confidence:** 4

**Summary:**

This work aims at improving the sample efficiency in terms of expert demonstrations in IL. In order to do so, it transforms the usual max-min objective of IRL into a single maximization objective by expressing both the reward and the policy in terms of the Q-function. The methods are supported with proofs and tested on both discrete and continuous action environments for the online setting and on simpler environments for the offline setting.

**Limitations And Societal Impact:**

yes

**Main Review:**

2. Strong points

The method seems well supported by the theoretical derivations and the experiments. It is tested on many different settings (offline/online, continuous/discrete actions, large/small observation space) and outperforms the baselines. It can also be used for IRL as it retrieves rewards close to the ground truth ones (at least in simple environments). The method seems to be implemented easily which would make it practical.


3. Weak points

   1- The related work section could be improved. To my knowledge, Klein et al. (2012) were the first to propose to represent the policy in terms of the Q-value to directly learn a reward function in Off-line IRL. Also, it would be interesting to discuss more in-depth the relationship of the proposed method with ASAF which also proposed an analytical solution to the maximization step of the min-max IRL objective enabling to directly learn the Q-values parametrizing the policy.

   2- The derivation of the method could be made clearer, in particular how to get to Eq. (5) and how it yields the objectives of the method Eqs. (9) and (10).


4. Recommendation

I recommend accept because the method seems efficient, practical, and well supported both theoretically and experimentally but I think the paper can be improved in terms of clarity and discussion with related methods. Additionally, some points remain to be clarified (see 5. Additional Remarks)

------------------------------
 ####  UPDATE
The author's response addressed my concerns and provided additional experiments so I decided to raise my score.

------------------------------

5. Additional Remarks

   1- In the continuous case, when SAC is used to improve the policy, the method falls back to a min-max objective. How do you explain that it does not experiment with the same drawbacks as adversarial methods such as ValueDICE and GAIL?

   2- I would be interested in seeing how the performance on Atari varies in terms of the number of expert demonstrations (like Fig 2.).

   3- I would be interested in seeing the performance of agents trained on the learned IRL reward as correlation is not always a good indicator of the quality of the reward. For example, the hardcoded reward of SQIL will train good agents and the one of GAIL can't (it is an apprenticeship algorithm and not an IRL one) yet SQIL has a lower correlation than GAIL (line 723 Appendix D).

   4- The methods you compare against on Atari were proposed for low dimension observations (MuJoCo), are there other IL/IRL methods you could compare against that were proposed for high dimensional envs?


6. Minor Remarks

   1- I am confused at line 490 and line 82: if $Q=\mathcal{B}Q$ and $Q=\mathcal{B}r$ then we would have $Q=r$ which does not make sense. Moreover $\mathcal{B}$ operates on $Q$ not $r$

   2- line 81-82, $Q$ is unique provided that $r$ is given but also $\pi$. It might be nice to differentiate $Q$ and optimal $Q$ (using for example $Q^*$ in Eqs. (1), (2), etc.)

   3- line 271-272 "with role similar to $r$" is confusing in the case of ASAF as the discriminator is not used to learn a reward in order to train the imitation policy like GAIL or AIRL (or the hard-coded reward of SQIL)


References

Klein, Edouard, et al. "Inverse reinforcement learning through structured classification." NIPS 2012. 2012.

**Time Spent Reviewing:**

4,5 hours

---

> ### Author Response · Authors · 2021-08-11
> **Additional Clarifications**
>
> **Comparison with adversarial IRL.**
> Main drawbacks of adversarial IRL methods like GAIL and ValueDICE are the extreme sensitivity to hyperparameters and training instability caused by adversarial optimization of a single objective. Instead, our method is non-adversarial as it relies on alternate gradient descent on two *separate* objectives for $\pi$ and $Q$ (section 4.4) that are non-competing. This is visualized in Fig. 1, where for a fixed $Q$,  SAC policy update is used to approximate a policy close to the $\pi_Q$ manifold and then an update step is taken for $Q$ using $\mathcal{J}$ along the manifold to reach the optimum.
>
> Alternatively, our method for the continuous case can be seen as an actor-critic method like SAC with a new critic rule, and shares methodology with a prior non-adverserial IRL method O-NAIL, as pointed by reviewer HuRr, which similarly optimizes $Q$ followed by a soft-actor-critic update and the same arguments from it hold for ours.
>
> **Atari: Performance with demos.** We will add additional figures showing atari performance with the number of expert demos to the supplementary. We show the rewards obtained below for two Atari environments vs # of demos:
>
> *Pong*
>
> | Num Demos | Rewards |
> | ---- | ----- |
> | 1 |  -3.2 |
> | 5 |  10.1|
> | 10 |  17.6|
> | 20 |  19.8 |
>
> *SpaceInvaders*
>
> | Num Demos | Rewards |
> | ---- | ----- |
> | 1 |  402.1 |
> | 5 |  540.3 |
> | 10 |  620.8 |
> | 20 |  820.3 |
>
> **Learned IRL reward.** We agree that correlation is not always a good indicator of the quality of the reward and experiment on the hopper environment using a single demo and find that using explicit recovered rewards obtains a performance of 3450.2.
>
> Nevertheless, we were not clear by what is meant by SQIL reward trains good agents and GAIL does not. SQIL simply gives 0 rewards to policy states and 1 for expert states for imitation learning, but is not an IRL method and does not recover any rewards that can be used for training an agent using normal RL training. For measuring the reward correlation with SQIL, we use the inverse bellmann equation to calculate a reward from $Q$, but find that this reward is not good and is not very correlated with the original env rewards.
> Whereas, although GAIL is an apprentiship learning method, it is formulated to find a reward function in the inner loop, such that the agent trained using the reward is close to the expert behavior. Thus the reward recovered using GAIL at the saddle point, should be able to train a good agent using RL. Our method in principle accomplishes the same thing but solves for an implicit reward function that trains a good agent.
>
>
> **Atari Comparisons.** Unfortunately, we are not familiar with IL/IRL methods other than BC and SQIL which have been shown to get good performances on Atari environments in literature.

---

> > ### Comment · Reviewer_rfsf · 2021-08-25
> > **Thank you for the response**
> >
> > The author's response addressed my concerns and provided the requested additional experiments so I decided to raise my score.

---

> ### Author Response · Authors · 2021-08-11
> **Author Response**
>
> Thanks for your time and effort! We hope the clarifications below sufficiently address your concerns and improve the understanding and clarity of our work. We will address the Additional remarks (point 5.) in separate comments. Please let us know if you may have any additional questions or comments, and we will try our best to allay them.
>
> **Related work.**
> Thank you for pointing out (Klein et al. 2012), which we were not familiar with. We will include this in the next revision of the related work, subject to space constraints. The idea behind the approach--learning a Q-value to simultaneously learn a policy and reward--is quite similar, although this earlier work only deals with hand-crafted state features and a discrete action space.
> We will also discuss ASAF further in the next revision. As we very briefly describe in the paper, it’s one of several methods that attempt to simplify the min-max IRL problem by fixing one of the components (in the case of ASAF, the reward function corresponding to the discriminator in GAIL) to have a certain form. This allows ASAF to minimize the JS divergence between expert and learned policy. A definite improvement over SQIL, which uses a fixed constant reward, ASAF still has some heuristic elements, such as the approximation of the expert density with policy densities due to the lack of access to the expert density. Our method can minimize many statistical distances (including the JS divergence) and avoids the approximation ASAF makes.
>
> **Derivation clarity.** We will improve the clarity of our derivations in the revision and any suggestions are greatly appreciated! Here, Eq. (5) is derived from the definition of $J(\pi, Q)$ in Lemma 3.2 and on simplification of the terms. The choice of regularizers $\psi$ in Eq.(6) and choosing a $\phi$ corresponding to them (Section 4.1) allows substituting $\psi$ in Eq. (5) to get Eq. (10). Eq. (9) similarly is derived for $J^*(Q)$ by setting $\pi$ to be $\pi_Q$ in $J(\pi, Q)$.
>
>
> **Typos.** Thanks for pointing out the typo in line 490. We denote the bellmann operator as $\mathcal{B}^\pi_r$, making the dependency of the reward explicit. Indeed, $\mathcal{B}^\pi_r$ operates on $Q$, and the statement should be modified to $Q = (\mathcal{T}^\pi)^{-1} r = \mathcal{B}^\pi_r Q$. Similarly, Lemma 3.1 should state for any $r$, $(\mathcal{T}^\pi)^{-1} r $ is the unique contraction of $B^\pi_r$. We will add the corrections in the revision.
>
> **Notation.** We will incorporate the suggestions, and make the notation in line 81-82 clearer in the revision.

---

### Official Review · Reviewer_HuRr · 2021-07-14

**Rating:** 9
**Confidence:** 5

**Summary:**

The paper presents an IL/IRL method that avoids estimating the reward during optimization by alternating between updates of the policy and Q-function instead, which builds on recent insights from ValueDice. Indeed, the optimization problem for learning the Q-function is very similar to the one used by ValueDice, when optimizing the reverse KL. Notable differences compared to ValueDice are the generalization to a large class of divergences (including f-divergences), non-adversarial formulation, and a procedure to recover rewards.
In particular, I think that the modification for recovering a state-only reward function is very interesting.
The algorithm only requires mild modifications of SAC (mainly by using a different loss for the Q-function) and shows promising results in experiment evaluations.

**Ethical Concerns:**

The paper does not raise ethical concerns.

**Limitations And Societal Impact:**

limitations and societal impact have been sufficiently addressed.

**Main Review:**

Quality
======
Overall, the approach seems technically sound and promising. Section 3 establishes the foundations for framing inverse reinforcement learning in terms of the Q function.  Section 4 exploits the insight that we do not need to reverse the order of $\text{IRL} \circ \text{RL}$ (which was proposed by GAIL) to make the optimization tractable, if we directly learn the Q-function instead of the reward function, since the Q-function is already an implicit representation of the optimal policy, making the inner reinforcement learning trivial.

The second important important insight is that the Q-function for a state-only reward function can be learned, simply by replacing the expectation over expert actions with an expectation over agent actions, which can be shown using importance sampling.

However, I have a question regarding the state-only rewards. It is not clear how a reward function of the form $r(s)$ is derived from the learned Q-function. From what I understand the learned models are all function of states and actions, and while the inverse Bellman update should result in a state-only reward, it is not clear to me how this step can be performed in practice.

Novelty / Significance
===================
* The presented derivations and algorithm are novel and very interesting. In particular, the derivation of the loss for the Q-function for general divergences and the modification for state-only rewards seem significant. Furthermore, the experimental results are promising.

* I think that Section 3 in itself does not contain that many new insights because it well known that the advantage function is a shaped [1] reward function, which also holds for MaxEnt RL, where maximizing the expected advantage + entropy regularization directly corresponds to minimizing the RKL to the distribution specified by A (or Q when disregarding normalization). Also, the transformation from the reward objective to Q-objective via the "inverse" Bellman operator strongly resembles the change-of-variables applied by DualDice/ValueDice. However, applying these ideas to Eq. 3 is an important contribution of the current submission.

* The idea to simplify IRL/IL by directly learning the Q-function is not novel, and was introduced already in the early times of IRL (e.g. Opt-Q [2]) and became more traction with recent deep learning methods such as ValueDice.

* The Q-update for the RKL looks very similar to the ValueDice objective, which I think should be stated more clearly. From what I can tell, the differences to ValueDice are 1) The use of the f-gan-formulation of the RKL instead of Donsker-Varadhan (e.g. using the expectation instead of the log-expectation in the first term), 2) Learning the soft-Q-function rather than the Q-function, 3) the actor update. Here 1) is rather insignificant as this option was also stated by Kostrikov et al. 2020. More similar is the connection to O-NAIL [3], which also uses the (DV or F-Gan) objective for the soft-Q-function in combination with the soft-actor-critic update.  Here the only difference seems to be that they provide an additional reward based on the last policy to incur a KL penalty between updates.


Clarity
======
The presentation of theory and algorithm is excellent.


References
==========
[1] Ng, Andrew Y., Daishi Harada, and Stuart Russell. "Policy invariance under reward transformations: Theory and application to reward shaping." Icml. Vol. 99. 1999.

[2] Dvijotham, Krishnamurthy, and Emanuel Todorov. "Inverse optimal control with linearly-solvable MDPs." ICML. 2010.

[3] Arenz, Oleg, and Gerhard Neumann. "Non-Adversarial Imitation Learning and its Connections to Adversarial Methods." arXiv preprint arXiv:2008.03525 (2020).

**Time Spent Reviewing:**

4

---

> ### Author Response · Authors · 2021-08-11
> **Author Response**
>
> Thank you for your time and effort! We are honored that you liked our work and found it to be in the top NeurIPS submissions.
>
> We will do our best to incorporate your suggestions and make connections with the related work like ValueDICE and O-NAIL more clear in the revision.
>
> For learning state-only reward functions, as noted we have forced the algorithm to constrain the rewards to be the same in a fixed state for different actions by using importance sampling in Eq. (15).  While recovering state-only rewards, we want this to force $Q(s, a)$ to have a structure such that $Q(s, a) -\gamma \mathbb{E}_{s'} [V(s’)]$ is independent of the action. In practice, when evaluating the inverse bellman equation, actions may still have a small role, but they become largely decoupled from the reward as visualized in Fig. 5, Appendix A. In addition, we can sample multiple actions from the policy for the same state in Eq.15 to enforce this soft constraint on the optimization.
>
> In future work, we will experiment more on the methodology stated in lines 463-466, which can make the constraint explicit by directly learning a $V(s)$ and eliminating actions.
>
> We will be glad to answer any other questions you may have. Thanks again!!

---

> > ### Comment · Reviewer_HuRr · 2021-08-20
> > **Re: Author Response**
> >
> > Thank you for your clarification, I think you should make the limitation of not directly learning a state-only reward function more clear in your paper and discuss how r(s) should be recovered. However, it does not affect my initial review, since I already assumed that the state-only reward function can not be recovered directly.

---

### Official Review · Reviewer_3TJb · 2021-07-16

**Rating:** 8
**Confidence:** 4

**Summary:**

This paper suggests an alternative approach, equivalent to adversarial IRL (that learns both the policy and the reward in an adversarial fashion), but in a direct way (learning the Q-function that encompasses both the reward and the policy). As in adversarial methods (GAIL, AIRL etc) they use the maximum entropy assumption and more explicitly, they assume that the optimal policy is greedy wrt an optimal soft-Q-function.

Using this relation between the optimal policy and the Q-function, they restrict the set of solutions to a manifold in which the objective only depends on the Q-function. They just have to minimize this objective (ensuring on the side that the policy stays in the manifold, i.e. it stays greedy) to obtain an optimal policy and Q-function. Their solution, in theory equivalent to the ones from an adversarial setting, is expected to outperform them as they avoid the learning difficulties of adversarial learning. Optionally, they can recover a reward function from the Q-function by simply inverting the Bellman operator.


**Limitations And Societal Impact:**

This paper has no societal impact.

**Main Review:**

This paper proposes a novel IL/IRL algorithm, that can be applied both in online and offline settings, that is easy to adapt for continuous states/action spaces (approximating the value function within a separate critic network). The method is simpler to implement (close to SAC) and proved to converge.

I would argue to accept it, my only concern being the lack of experiments:
Mujoco: Did the authors try Humanoid?
Atari: only 3 games, I understand that these environments are longer to train and require more engineering, but presenting just 3 envs looks like they tried more but had less good results on the other. I believe this is not the case, but having more games would really improve the paper (even if the results are not that good in other games, the paper is already great).
Reward reconversion: having results from more environments (at least the other mujoco) would be great to convince that the reward is correlated.

However, the theory part is nice enough and the already given results are promising, so I will go for an accept. But if the authors can provide more results in the paper revision I will even increase my score.

Details:
A closely related IRL work inverses soft-policy improvement also to recover the soft-Q-function, but assuming the observed trajectories are generated by a soft-Q learner and not an expert:
Learning from a Learner (ICML 2019) by Jacq, Geist, and Pietquin.

In Eq.1, the notation $\pi^*$ is confusing, as it should be used for the policy that is greedy wrt the optimal soft-Q function ($\pi^*(a\vert s) \propto \exp(Q^*(s,a))$).

Same for $J^*(Q)$, that would be the optimized objective after convergence.


**Time Spent Reviewing:**

5

---

> ### Author Response · Authors · 2021-08-11
> **Author Response**
>
> Thanks for your time and effort. Great suggestions and comments. We were encouraged that you liked our paper and do our best to address your concerns below:
>
> **Related work.** Thanks for pointing out the ‘Learning from a Learner' work. We were not aware of it, but it is very interesting and indeed in a similar vein and we will discuss it in related works, space permitting.
>
> **Notation Fixes.** We will improve the clarity and try to incorporate the suggested fixes in the next version.
>
> **Additional experiments.** ​​ In the submission, for the sake of time and brevity, we show results on a set of environments focused on by prior works such as SQIL and ValueDice. We report additional results on the Humanoid task and three additional Atari environments for which pre-trained experts were readily available using  [RL Baselines3 Zoo](https://github.com/DLR-RM/rl-baselines3-zoo) (we follow the same experimental settings as the main paper). We will add these additional results in supplementary.
>
> *Humanoid*:
>
> | Ours  |  Expert |
> | ----------- | ----------- |
> | 5227.1 | 5312.8|
>
>
> *Atari*:
>
> | Env | Ours  |  Expert |
> | ----------- | ----------- | ----------- |
> | BeamRider | 3025.4 | 4295.9 |
> | Seaquest |  2349.6 | 2393.3 |
> | Qbert | 12940.0 | 11496.8 |
>
> **Reward correlations.** Due to the short response window, we were only able to calculate reward correlations with three more environments shown below. We will include a complete table on all the tested Mujoco envs in the next version.
>
>
> | Env | Reward Correlations |
> | ----------- | ----------- |
> | LunarLander |  0.92 |
> | Half-Cheetah | 0.86 |
> | Pong | 0.67 |

---

> > ### Comment · Reviewer_3TJb · 2021-08-25
> > **Thanks for the additional results.**
> >
> > As promised, the authors addressed my concerns and provided more experimental results that confirm the strength of their approach, hence I increase my score.

---

### Official Review · Reviewer_XfMp · 2021-07-16

**Rating:** 7
**Confidence:** 4

**Summary:**

This paper proposes to get around the typical two-stage inverse RL pipeline of (1) learn a reward model and (2) optimize it, by encapsulating both rewards and dynamics in a single Q function, learned by imitation. This reframing of IRL preserves the standard properties of IRL but is simpler to optimize and avoids the need for adversarial training used by other methods. This results in a simple imitation learning method that is competitive with or outperforms reasonable baselines on a few standard benchmarks. The proposed algorithm has other nice features, including the ability to easily recover the learned reward and to be trained both offline and online, and it can be easily rephrased to obtain a method for imitation from observations.

**Limitations And Societal Impact:**

Yes.

**Main Review:**

Positives:
- See my summary. I found the core technical contributions of this work to be novel, and potentially high-impact, as they simplify and reframe IRL to remove a common annoyance: the need for two-stage (or simultaneous two-part) training.
- The theoretical analysis is correct as far as I can tell. Although certain details were overemphasized to my taste (e.g. the choice of regularizers, which didn't make a huge difference empirically), the exposition was quite clean and easy to follow.
- The experiments compare against a representative set of reasonably strong baselines, and include enough domains to suggest that this method will work generally.

Negatives:
- As far as I can tell, none of the experimental domains here require generalization. These domains include very stereotyped initial state distributions, which means that an imitation agent that exactly replays the expert demonstration will solve the task. This likely plays a role in the proposed method's ability to perform well given a single demonstration. And in my opinion it is misleading, because it sidesteps one of the principle challenges in RL and IL, namely learning policies that can be used in the presence of at least mild occupancy drift (see [1] and [2] for discussions). Ideally, imitation learning methods should be evaluated in settings where the expert initial state distribution does not completely cover the states the agent can start in, such as the DeepMind Control Suite (as discussed in [2]), procedural tasks like CoinRun (as used in [3]) or tasks with more challenging perception (such as the door-opening task in [4]).
- The proposed baselines are not as strong as they could be. In particular, Discriminator-actor-critic [5] or strongly regularized GAIL variants (such as [6]) are closer to what are used in practice than vanilla GAIL and typically give much better results. And another recently published method [4] produces similar results on Gym with single demonstrations, but is not mentioned or compared to.

Minor:
- I found the point made in footnote 2 to be obscure as phrased. Can you rephrase this to explain exactly how this relates to (6) and what this has to do with arbitrary experts?

Despite the limitations of the experimental evaluation, I believe the proposed algorithm is a solid and potentially quite impactful contribution to the IRL and imitation learning literature. I found the exposition insightful and I admire the simplicity of the proposed result.

I am willing to raise my score if the limitations of the experimental evaluation are addressed.

[1] Mania et al - NeurIPS 2018 - Simple random search of static linear policies is competitive for reinforcement learning

[2] Jaegle et al - ICML 2021 - Imitation by Predicting Observations

[3] Edwards et al - ICML 2019 - Imitating Latent Policies from Observation

[4] Dadashi et al - ICLR 2021 - Primal Wasserstein Imitation Learning

[5] Kostrikov et al - ICLR 2019 - Discriminator-Actor-Critic: Addressing Sample Inefficiency and Reward Bias in Adversarial Imitation Learning

[6] Peng et al - ICLR 2019 - Variational Discriminator Bottleneck: Improving Imitation Learning, Inverse RL, and GANs by Constraining Information Flow

**Time Spent Reviewing:**

5

---

> ### Author Response · Authors · 2021-08-10
> **Generalization concerns**
>
> **Generalization over initial distribution shift**. We perform additional experiments over two different settings: 1) Modified LunarLander env motivated by [1] and 2) DeepMind Control Suite as suggested:
>
>
> ***1) LunarLander with initial shift***
>
> We experiment with initial shift distribution in the LunarLander-v2 environment similar to [1]. The agent is typically initialized in a small zone at the middle top of the screen. Instead, we modify the environment to initialize the agent near the top-left corner of the screen. We use experts from the unmodified environment, and test whether the agent can still learn to land the lunar lander while recovering from the initial distribution shift.
>
> **Results**
>
> *Offline Case*: We find in the offline case that the agent cannot learn to recover from the occupancy shift. The lander typically tends to fly off the frame and shows random behavior. This is expected as the IQ-learn algorithm is not aware of the shift of initial distributions of the agent and the expert, and can’t explore the environment to correct the initial state shift to match the occupancy distributions.
>
> *Online Case*: Whereas in the online case, we find that the agent can sufficiently explore the environment, and learns a behavior of first horizontally moving the lander from the top left to the top center and then successfully imitating the original expert trajectory, receiving an avg. episode reward of ~250 with 10 expert demos.
>
> An extra consideration here is in Eq. 9, where we originally only apply reward regularization to the expert states, but we find applying regularization to both expert and policy states to be beneficial in this case. As it enforces the learning of an implicit reward function that can generalize outside the expert distribution to more arbitrary policy states.
>
> ***2) DeepMind Control Suite***
>
> We only experiment with the *reacher_easy* task in DeepMind Control Suite for the sake of time but are willing to add more experiments to the revised version on request. We choose the reacher environment as the target position is randomly initialized in each episode, making it a good setting for testing generalization for imitation learning.
>
> **Results**:
>
> We test with different number of expert demonstrations: 1,5, 10, 20, each with different target positions on the offline and online settings (we report averaged results over 100 episodes with different targets). The avg. expert performance is ~990 in this case.
>
> *Offline Case*: In the offline setting, a single demonstration is typically not enough to learn a generalized reward function and leads to a reward that overfits to a particular target position. We quantify the results below, with the observation that imitation learning performance improves with # of expert demos. This can be justified, as more experts with different targets allow learning a reward function that is better generalizable.
>
> | Num Experts | Rewards |
> | --- | ----------- |
> | 1 | 105.4 |
> | 5 | 120.1 |
> | 10 | 210.6 |
> | 20 | 325.0 |
>
>
> *Online Case*: In the online setting, the algorithm is able to explore the environment over different episodes and can learn to correct the behavior leading to better performance.  In particular, given a sufficient number of expert demos, it can learn to associate what expert behavior to imitate given a particular target and learns a more general reward function. We show quantitative results below:
>
> | Num Experts | Rewards |
> | --- | ----------- |
> | 1 | 271.3 |
> | 5 | 485.1 |
> | 10 | 545.0 |
> | 20 | 734.9 |
> | 50 | 926.1 |
>
>
> [1] Reddy et al - ICLR 2020 - SQIL: Imitation Learning via Reinforcement Learning with Sparse Rewards

---

> > ### Comment · Reviewer_XfMp · 2021-08-14
> > **Thanks for the clear response, a few additional questions**
> >
> > Thanks to the authors for the detailed reply and for providing these additional experiments. The reply addresses most of my concerns, but I think the following points could benefit from additional clarification. In the context of the replies provided, I still maintain my original rating of Accept.
> >
> > **Lunar lander experiments**
> >
> > (1) "An extra consideration here is in Eq. 9, where we originally only apply reward regularization to the expert states, but we find applying regularization to both expert and policy states to be beneficial in this case."
> >
> > Can the authors please provide the full expression that's optimized in this case? I'm not sure I understand exactly what's changed in this form of regularization, which isn't used in other experiments.
> >
> > (2) “but we find applying regularization to both expert and policy states to be beneficial in this case.”
> >
> > Can the authors please provide a quantitative evaluation of this claim? Regularization of this form isn't used for any of the other experiments in the paper, as far as I can tell, and this data would be very helpful for evaluating how important this regularization is to producing generalization away from expert states.
> >
> > **Control Suite Reacher experiments**
> >
> > (1) Is the regularizer described for the Lunar lander experiments above also used for the Control Suite experiments? If not, can the authors provide an explanation for why it's needed to obtain generalization in Lunar lander but not in reacher easy?
> >
> > (2) In my opinion, these experiments do suggest that IQ-learn can indeed generalize appropriately. But given that it takes ~50 demos and online experience to reach performance comparable to the expert on a domain as simple as reacher_easy, I believe the authors should tone down and carefully reconsider claims such as the following throughout the paper. “In our experiments, we find that our method is performant even with very sparse data - surpassing prior methods using one expert demonstration in the completely offline setting". Performance comparable to the expert with only a single demonstration is observed only for domains where no generalization is required. I believe this caveat makes the claim very misleading, given that in practice it is uncommon for the expert and imitator initial states to match.

---

> > > ### Author Response · Authors · 2021-08-21
> > > **Additional Clarifications**
> > >
> > > **Lunar lander experiments**
> > >
> > > (1) Hi, we are optimizing Eq. 9 using the X^2 regularization in this case (as given in Section 5.3). This adds an L2 penalty on rewards only in expert states given by the 3rd term of the expression. In the case of initial distribution shift, where the learned policy significantly differs from the expert distribution, adding L2 penalization on rewards seen in policy states improves the generalization of the learned reward functions to states unseen by the expert.
> > >
> > > (2) Using 10 experts on the modified lunar lander environment, without regularization we get avg. rewards ~205, and with regularization ~250. Our core focus in the work is to improve (basic) imitation learning, and we don't target domain shift as the authors felt it might be better addressed in separate follow-up work (rather than an extra appendix). For the experiments in the main paper, we didn't find a significant gain using the L2 penalization on policy states and omit it for simplicity.
> > >
> > >
> > > **Control Suite Reacher experiments**
> > >
> > > (1) Yes, sorry for not specifying, but we also use the additional L2 penalization on the policy states in this case.
> > >
> > > (2) We’re glad that the additional experiments are convincing to show that IQ-learn can generalize well. On the subject of performance, we stress that *reacher_easy* is considered a very challenging environment to solve with imitation learning. The reason is that, unlike other single-task environments, this environment is actually multi-task with the goal-state changing every episode randomly within a large circular region. Such environments have been found to be very difficult to solve using IRL [1] as a large number of expert demos are needed to fully cover the goal distributions, and usually require meta-IRL methods that can figure a task context like PEMIRL [1]. We tried BC and GAIL on *reacher_easy* and found even with 50 experts they obtain mean rewards of 325.2 and 440.1 respectively, which is equivalent to what we see using our method with just 5 expert demos! It is surprising to us that the method can learn a reward to figure out what goal state to reach, even if it is not engineered specifically to do so.
> > > We will qualify our statement about single demonstration with a discussion of when this is possible due to experts having to cover the support of the optimal policy.
> > >
> > > [1] Yu et al - NeurIPS 2019 - Meta-Inverse Reinforcement Learning with Probabilistic Context Variables

---

> ### Author Response · Authors · 2021-08-10
> **Author Response**
>
> Thanks for your time and effort. Nice suggestions and comments. We are delighted that you found the work insightful and of high impact. We address generalization concerns in separate comments and put clarifications for other concerns below. We would love to answer follow-up questions if any concerns are not addressed. Thanks again!!
>
> **Fixes.**  We will rephrase footnote 2 to be more clear. In principle, the idea is that using a set of regularizers that take expectation over the expert distribution instead of a fixed distribution as in Eq. 6, allows dealing with different expert behavior and easily accommodates multimodality.
>
> **Baselines.**  In the submission for sake of brevity and time, we found ValueDICE to be the strongest prior work to compare against and skipped other recent baselines for presenting results. Based on the suggestions we try additional baselines:
>
> For [4], we found that the authors don't provide their code, making it hard to compare against, although we will add it in our related works.  Their results are comparable with Discriminator-actor-critic (DAC), and we add it as an additional baseline to compare against:
>
> **DAC Results**:
>
> - For Atari, we found that the method fails to scale similar to ValueDICE/GAIL and performs near-random in 1M env steps.
>
> - For Mujoco envs, we include a table of baseline results using a single expert demo:
>
> | Env      | Rewards |
> | ----------- | ----------- |
> | Hopper      | 3305.1       |
> | Half-Cheetah   | 4080.6        |
> | Walker |  4107.9  |
> | Ant | 1437.5 |

---

### Public Comment · ~Minghuan_Liu1 · 2021-12-01
**Great paper but is there a misplacement in figure 1?**

First thanks to the author for such a great work. However, I found a little confused when reading figure1, where it says $J(\pi, \cdot)$ is concave and $J(\cdot, Q)$ is quasi-convex, however, this seems to be in conflict with the proof shown in Lemma C.1 and Lemma C.2? Also this does not coincide with the x and y-axis shown in fugure 1.

---

> ### Public Comment · Authors · 2021-12-29
> **Author Response**
>
> Hi Minghuan, we are glad you liked our work. Thanks for bringing out this point, the notation here can be misleading. \
> For $\pi \in \Pi$ and $Q \in \Omega$, $J(\pi, \cdot)$ denotes a function in $Q$, mapping from $\Omega \rightarrow \mathbb{R}$ for fixed $\pi$.
> Similarly, $J(\cdot, Q)$ denotes a function in $\pi$, mapping from $\Pi \rightarrow \mathbb{R}$ for fixed $Q$.
>
> We believe that Figure 1 is correct but we, unfortunately, made this typo in the notations for Lemma C.1 and Lemma C.2 and have fixed it in our revision to camera-ready. Thanks again for catching this and we hope that clarifies any confusion.

---

### Decision · Program_Chairs · 2021-09-27

**Decision:**

Accept (Spotlight)

**Comment:**

This paper received very positive reviews initially. The reviewers liked the originality of the work, it's theoretical soundness and the good performance of the algorithm. They had some concerns and wanted to see more about generalization capacities, stronger baselines and a larger set of test environments. The discussion was rich enough to convince the reviewers to raise their scores. The authors made additional experiments and addressed all the concerns of the reviewers.